# Project and Probe: Sample-Efficient Adaptation by Interpolating Orthogonal Features

**Annie S. Chen**[*1]**, Yoonho Lee**[*1]**, Amrith Setlur**[2]**, Sergey Levine**[3]**, Chelsea Finn**[1]

Stanford University[1], Carnegie Mellon University[2], UC Berkeley[3]

`asc8@stanford.edu, yoonho@stanford.edu`

## Abstract

Transfer learning with a small amount of target data is an effective and common approach to adapting a pre-trained model to distribution shifts. In some situations, target data labels may be expensive to obtain, so we may only have access to a limited number of target data points. To make the most of a very small target dataset, we propose a lightweight, sample-efficient approach that learns a diverse set of features and adapts to a target distribution by interpolating these features. Our approach, Project and Probe ($PRO^2$), first learns a linear projection that maps a pre-trained embedding onto orthogonal directions while being predictive of labels in the source dataset. The goal of this step is to learn a variety of predictive features, so that at least some of them remain useful after distribution shift. $PRO^2$ then learns a linear classifier on top of these projected features using a small target dataset. Theoretically, we find that $PRO^2$ results in more sample-efficient generalization by inducing a favorable bias-variance tradeoff. Our experiments on four datasets, with multiple distribution shift settings for each, show that $PRO^2$ improves performance by 5-15% when given limited target data compared to prior methods such as standard linear probing.

## 1 Introduction

Machine learning models can face significant challenges when there is a distribution shift between training and evaluation data. A model trained on a specific source dataset may not perform well when deployed on a target domain with a distribution of inputs that differs significantly from the source domain. One common and reliable approach for adapting to distribution shifts is fine-tuning a trained model on a small amount of labeled data from the new target domain. However, in some situations, target data labels may be expensive to obtain, which limits the number of available labeled datapoints for fine-tuning. As an example, a hospital may have imaging software that slightly differs from what was used for dataset collection, but they may not be able to acquire many new labeled samples. In such conditions, conventional fine-tuning approaches may overfit to the small target dataset and distort the information learned during initial training. Therefore, we require a method that can reliably extract information from the new target domain with less overfitting.

Recent works have demonstrated the effectiveness of re-training a final linear head using target data for adapting to distribution shifts due to spurious correlations or domain shift (Rosenfeld et al., 2022; Kirichenko et al., 2022; Mehta et al., 2022). However, it is unclear whether this standard approach of re-training a linear layer is the most data-efficient method to adapt pre-trained features to various target distributions. While versatile, feature embeddings may not necessarily contain the most suitable set of features for adapting to target distributions: they may also contain redundant, non-predictive, or noisy information. Our primary insight is that the key to more sample-efficient adaptation to target domains lies in starting with a compact and diverse set of useful features. Each feature in this set should not only be predictive, but also hold unique information distinct from others inside the set. We leverage source data, which is substantially more abundant than target data, in performing this selection of features for target adaptation.

We propose Project and Probe ($PRO^2$), a simple and sample-efficient method for adapting to unknown target distributions. $PRO^2$ first learns a projection of pre-trained embedding vectors, which is optimized to extract a diverse set of features that are each predictive of labels. More specifically,

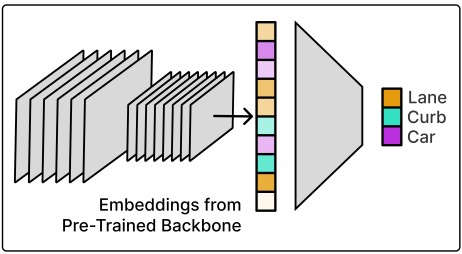
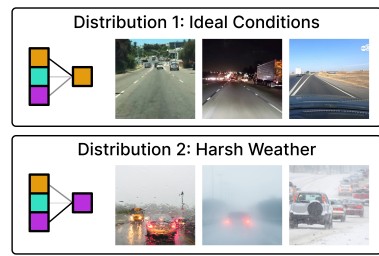

(a) **Pro**ject with Large Source Dataset  (b) **Pro**be with Small Target Dataset

Figure 1: **The Project and Probe (PRO$^2$) framework for adapting to different target distributions.** (a) We first use a large source dataset to project pre-trained feature embeddings onto a set of predictive features while enforcing orthogonality. (b) For a new target distribution, we learn a linear layer on top of the projected features. This step adaptively chooses features in a data-efficient manner.

we first use a source dataset to project pre-trained feature embeddings onto a smaller set of predictive features. We enforce pairwise orthogonality among all features, thereby ensuring that each projected dimension carries unique information not present in others. We expect this learned feature space to compactly contain a diverse set of predictive features while discarding information that is redundant or not predictive on the task. PRO$^2$ then uses the reduced set of features as a basis space for adaptation. Specifically, we fit a linear head on top of the projected embedding using labeled target data. Both the linear projection and the linear head require minimal computational overhead, making PRO$^2$ a practical method for adapting to new target distributions. Fig. 1 shows a visual summary of PRO$^2$.

To support our approach, we provide a theoretical analysis, in both a general setting with minimal distribution assumptions as well as the more specific setting of a shifted homoscedastic Gaussian model, showing how PRO$^2$ learns a projection matrix that results in better generalization due to a favorable bias-variance tradeoff. From this analysis, PRO$^2$ improves sample efficiency because it can learn useful, diverse features so that it is more likely to better recover the important directions for adaptation with a smaller projection dimension, allowing us to combat the variance introduced by a very small target dataset while maintaining low bias. We conduct experiments on a variety of distribution shift settings across 4 datasets. We find that standard linear probing, which is the default method used by prior works, is not the most data-efficient adaptation approach. Using PRO$^2$, i.e. projecting with source data onto an informative feature-space basis and probing with target data, improves performance by 5-15% in few-shot adaptation to new target distributions.

## 2 RELATED WORK

**Robustness and zero-shot generalization.** Many prior works aim to improve robustness to various distribution shifts (Tzeng et al., 2014; Ganin et al., 2016; Arjovsky et al., 2019; Sagawa et al., 2020; Nam et al., 2020; Creager et al., 2021; Liu et al., 2021; Zhang and Ré, 2022). Additionally, prior works have studied how to adapt pre-trained features to a target distribution via fine-tuning Oquab et al. (2014); Yosinski et al. (2014); Sharif Razavian et al. (2014). Such fine-tuning works typically frame robustness to distribution shift as a zero-shot generalization problem Kornblith et al. (2018); Zhai et al. (2019); Wortsman et al. (2022); Kumar et al. (2022), where the model is trained on source and evaluated on target. Both of the above classes of approaches fundamentally cannot handle the problem settings we consider, where a single function is insufficient for achieving good performance on different distributions. In this paper, we evaluate on a variety of test distributions, some of which are mutually exclusive, and it is therefore crucial to perform adaptation on the target distribution.

**Adapting to distribution shifts.** Recent works have proposed various methods for adapting models at test time with some labeled target data Sun et al. (2020); Varsavsky et al. (2020); Iwasawa and Matsuo (2021); Wang et al. (2020); Zhang et al. (2021); Gandelsman et al. (2022); Lee et al. (2022a). In particular, given a feature embedding produced by a pretrained network with sufficient expressivity, training a final linear head, also known as linear probing, suffices for adapting to datasets with spurious correlations Kirichenko et al. (2022); Mehta et al. (2022); Izmailov et al. (2022) as well as in the setting of domain generalization Rosenfeld et al. (2022). As detailed further in Sec. 3, we specifically focus on scenarios in which we have very little target data (only $4 \sim 256$ datapoints). We

find that in this setting, training a final linear head in the default manner is not the most data-efficient way to adapt. $\text{PRO}^2$, which breaks this training down into 2 steps, is able to more effectively extract useful features and interpolate between them for varying target distributions, leading to improved sample efficiency with limited target data.

**Learning diverse features for spurious datasets.** Prior works have explored learning diverse or orthogonal features in standard supervised learning settings (Bansal et al., 2018; Xie et al., 2017b;a; Laakom et al., 2023b; Cogswell et al., 2015; Laakom et al., 2023a; Zbontar et al., 2021), and we show diversification can lead to more sample efficient adaptation to distribution shifts. Neural networks tend to be biased towards learning simple functions that rely on shortcut features (Arpit et al., 2017; Gunasekar et al., 2018; Shah et al., 2020; Geirhos et al., 2020; Pezeshki et al., 2021; Li et al., 2022; Lubana et al., 2022). To better handle novel distributions, it is important to consider the entire set of functions that are predictive on the training data (Fisher et al., 2019; Semenova et al., 2019; Xu et al., 2022). Recent diversification methods for adaptation discover such a set (Teney et al., 2022; Pagliardini et al., 2022; Lee et al., 2022b). The latter two methods use additional assumptions such as unlabeled data. With a similar motivation to ours, Teney et al. (2022) penalizes the similarity between different features, but does so with an additional loss term instead of explicitly enforcing orthogonality. We observe that this implementation detail matters in Sec. 6, where $\text{PRO}^2$ outperforms Teney et al. (2022). A concurrent work (Morwani et al., 2023) also proposes an orthogonal projection method to learn diverse classifiers. However, the Probe step of $\text{PRO}^2$ additionally interpolates between the orthogonal features, and we provide theoretical and empirical analysis of how distribution shift severity affects sample efficiency during probing.

**Compression & feature selection.** In aiming to extract important features and discarding repetitive information, $\text{PRO}^2$ is related to work on compression May et al. (2019) and information bottlenecks Tishby et al. (2000); Alemi et al. (2016). Our method is also closely related to methods that learn projections such as principal component analysis (PCA) and linear discriminant analysis (LDA). Beyond these representative methods, there is an immense body of work on feature selection (Dash and Liu, 1997; Liu and Motoda, 2007; Chandrashekar and Sahin, 2014; Li et al., 2017) and dimensionality reduction (Lee et al., 2007; Sorzano et al., 2014; Cunningham and Ghahramani, 2015). Among all projection-based methods, LDA is the most related to ours, but it only learns the single most discriminative direction. In Corollary 9, we show that $\text{PRO}^2$ with dimensionality $d = 1$ provably recovers the LDA direction in a shifted homoscedastic Gaussian model, and that using higher values of $d$ is critical in adapting to higher degrees of distribution shift. Generally, most methods (including LDA) operate in the setting without distribution shift.

## 3 ADAPTATION TO DISTRIBUTION SHIFT

We now describe our problem setting, where the goal is to adapt a model so as to provide an accurate decision boundary under distribution shift given a limited amount of target distribution information. We consider a source distribution $p_S(x, y)$ and multiple target distributions $p_T^1(x, y), p_T^2(x, y), \ldots$. The source dataset $\mathcal{D}_S \in (\mathcal{X} \times \mathcal{Y})^N$ is sampled from the source distribution $p_S$. We evaluate adaptation to each target distribution $p_T^i$ given a small set of labeled target data $\mathcal{D}_T^i \in (\mathcal{X} \times \mathcal{Y})^M$, where $M \ll N$ so the model must learn from both the source and target data for best performance. We measure the post-adaptation average accuracy of the model on a held-out target dataset from the same distribution $p_T^i$.

We note that this setting differs from the setting studied in prior works on spurious correlations (Sagawa et al., 2020), which train a model only on source data $\mathcal{D}_S$ and evaluate the model's performance on the hardest target distribution (i.e., worst-group accuracy). This is also different from the setting used in fine-tuning methods for zero-shot generalization (Wortsman et al., 2022; Kumar et al., 2022): such methods fine-tune a pretrained model on source data $\mathcal{D}_S$ and directly evaluate performance on target data $\mathcal{D}_T^i$ without any exposure to labeled target data. Compared to these zero-shot evaluation settings, we argue that a small amount of target data may realistically be required to handle the arbitrary distribution shifts that arise in the real world. Target data can be an effective point of leverage because it can be available or easy to collect, and we find that even a small dataset can reveal a lot about what features are effective in the target distribution. Our problem setting of adapting with target data has been used in some recent works (Kirichenko et al., 2022; Rosenfeld et al., 2022; Izmailov et al., 2022; Lee et al., 2022a), but we specifically focus on the setting in which we only have access to a very small target dataset, i.e., $M \ll N$.

---

**Algorithm 1** Project and Probe

**Input:** Source data $\mathcal{D}_S$, Target data $\mathcal{D}_T$,
Backbone $f : \mathcal{X} \to^D$

Initialize $\Pi :^D \to^d$       **#Project** with source
**for** $i$ in $1 \ldots d$ **do**
    $\Pi_i \leftarrow \arg\min \mathcal{L}_S(\Pi_i(f(x)), y)$
        subject to $\Pi_j \perp \Pi_i$ for all $j < i$
**end for**

Initialize $g :^d \to \mathcal{Y}$      **#Probe** with target
$g \leftarrow \arg\min \mathcal{L}_T(g(\Pi(f(x))), y)$

---

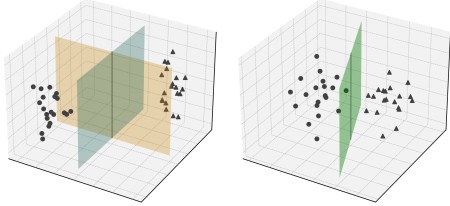

(a) Two orthogonal features    (b) Interpolated classifier

Figure 2: Visualization of PRO$^2$: (a) orthogonal decision boundaries learned during the Project stage, and (b) the interpolated classifier learned during the Probe stage.

## 4 PROJECT AND PROBE

We now describe PRO$^2$, a framework for few-shot adaptation to distribution shifts. PRO$^2$ is composed of two steps: (1) learn a projection $\Pi$ that maps pre-trained embeddings onto orthogonal directions, and (2) learn a classifier $g$ using projected embeddings.

Before Step (1), we use a pre-trained backbone model $f : \mathcal{X} \to \mathbb{R}^D$ to map the datapoints to $D$-dimensional embeddings. This backbone model extracts meaningful features from the raw inputs, resulting in a low-dimensional embedding space, for example $224 \times 224 \times 3$ images to $D = 1024$-dimensional embeddings.

**Step 1: Project with source.** Recall that we operate in the few-shot setting, where we may have fewer target datapoints than even embedding dimensions ($M < D$). Intuitively, we would like to select a suitable decision boundary from a set of decision boundaries that worked well in the source domain. If this set is discrete, that might correspond to training some sort of diversified ensemble of linear classifiers on top of the features, a strategy adopted in some prior works (Teney et al., 2021; Lee et al., 2022b; Pagliardini et al., 2022).

However, in general, we might need the expressive power of a continuous set of decision boundaries to adapt to the target domain, and we can construct this set by *interpolating* over a basis of decision boundaries. Mathematically, this is identical to selecting a set of linear features. Thus, the question we must answer is: which set of linear features of the $D$-dimensional feature space should we retain? First, it should be clear that the features should form an orthogonal basis, as otherwise they will be redundant. Second, the features should be discriminative, in the sense that they are sufficient to solve the desired prediction task. Lastly, there should not be too many of them, since the more features we include (i.e., the larger the rank of the basis we learn), the more samples we'll need from the target domain to find the best decision boundary in the corresponding set.

To learn a feature space that satisfies these desiderata, we parameterize a linear projection $\Pi : \mathbb{R}^D \to \mathbb{R}^d$ that maps the embeddings to a reduced space ($d \leq D$). Specifically, we use the source data to learn a complete orthonormal basis for the embedding space $\Pi_1, \Pi_2, \ldots, \Pi_d \in^D$, by learning each basis vector with the constraint that it is orthogonal to all vectors before it:

$$\Pi_i = \arg\min \mathbb{E}_{(x,y) \sim \mathcal{D}_S} \mathcal{L}(\Pi_i(f(x)), y) \quad \text{s.t.} \quad \Pi_j \perp \Pi_i \text{ for all } j < i. \tag{PRO$^2$}$$

Note that this induces a natural ranking among the basis vectors. This collection of orthogonal vectors constitute the rows of our projection matrix $\Pi$. In our implementation, we do projected gradient descent, enforcing orthogonality using QR decomposition on the projection matrix after every gradient step. See Appendix B for a short PyTorch implementation.

Empirically and theoretically, we find that it is particularly beneficial to use a small $d \ll D$, even $d = 1$, in when adapting to small distribution shifts and use larger $d$ for more severe distribution shifts.

**Step 2: Probe with target.** After learning $\Pi$, we learn a classifier $g : \mathbb{R}^d \to \mathcal{Y}$ that maps the projected embeddings to the target labels:

$$g = \arg\min \mathbb{E}_{(x,y) \sim \mathcal{D}_T} \mathcal{L}(g(\Pi(f(x))), y).$$

Since the projection $\Pi$ was optimized to a diverse set of the most discriminative features for the source data, we expect the initial projected features to be particularly predictive when the distribution shift is relatively small.

In summary, PRO$^2$ is a simple and lightweight framework that addresses the problem of few-shot adaptation in the presence of distribution shifts. We summarize its overall structure in Algorithm 1 and show a simplified 3D visualization in Fig. 2. In our implementation, we use cached embeddings for all source and target datapoints, such that feeding raw inputs through $f$ is a one-time cost that is amortized over epochs and experiments, making our framework scalable and efficient. As an orthogonal improvement to our work, one could additionally fine-tune the backbone network on source data. In Sec. 5, we theoretically analyze the properties of the projection and classifier learned by PRO$^2$. We then empirically evaluate PRO$^2$ on a variety of distribution shifts and publicly available backbone networks in Sec. 6.

## 5 ANALYSIS

In this section, we present a theoretical analysis of PRO$^2$, aiming to understand how our proposed orthogonal feature selection procedure can lead to sample-efficient adaptation under distribution shifts. Intuitively, the more shift we can expect, the more features we should need to adapt to it, which in turn requires more samples during adaptation (to fit the features accurately). However, the choice of how we extract features determines how the sample complexity grows under distribution shift: while large shifts may still require many features, if the features are prioritized well, then smaller shifts might require only a very small number of features, and thus require fewer samples.

Given $M$ target samples, in our analysis we first show that using (fewer) $d$ features leads to lower variance, which scales as $\mathcal{O}(d/M)$, as opposed to $\mathcal{O}(D/M)$. But this improvement comes at a cost in bias, which in some cases scales as $\mathcal{O}(\sqrt{D-d} \cdot \text{KL}(p_S \| p_T))$. Note that this term grows with the amount of shift between the source $p_S$ and target $p_T$ distributions. In Sec. 5.1, we first analyze the specific features learned by PRO$^2$ with minimal distributional assumptions. Then, in Sec. 5.2, we apply our general results to a shifted homoscedastic Gaussian (SHOG) model, where the bias and variance terms take more intuitive forms. We also empirically verify our results using synthetic SHOG data. Additional theoretical results and proofs can be found in Appendix A.

### 5.1 BIAS-VARIANCE TRADEOFFS FOR GENERAL SHIFTS.

In this section, we analyze the properties of $\Pi$ (learned projection) on the target distribution to understand why sample efficiency during adaptation can be improved by first extracting a set of diverse features that are predictive on the source distribution.

**Probing on the target distribution.** We first introduce some additional notation specific to the target distribution. Let $\mathcal{S}_d$ denote $\text{span}(\{\Pi_i\}_{i=1}^d)$ and $\mathsf{M}_d$ denote the projection matrix for $\mathcal{S}_d$,

$$\mathsf{M}_d = (\Pi^\top \Pi)^{-1} \Pi^\top \quad \text{where,} \quad \Pi \triangleq \Pi_1, .., \Pi_d. \tag{1}$$

We denote the target error for classifier $\mathbf{w}$ as $\mathcal{L}_T(\mathbf{w}) \triangleq \mathbb{E}_{p_T} l(\langle \mathbf{w}, \mathbf{x} \rangle, \text{ y})$, and the bias incurred by probing over the projected features $\text{span}(\{\Pi_i\}_{i=1}^d)$ as:

$$b_d \triangleq \min_{\mathbf{w}' \in \mathcal{S}_d} \mathcal{L}_T(\mathbf{w}') \; - \; \min_{\mathbf{w} \in \mathcal{W}} \mathcal{L}_T(\mathbf{w}).$$

Finally, the $d$-dimensional weight vector learned by PRO$^2$ on the $M$ projected target samples is:

$$\hat{\mathbf{w}}_d \triangleq \arg \min_{\substack{\mathbf{w} \in \mathcal{S}_d \\ \|\mathbf{w}\|_2 \leq 1}} \sum_{i=1}^M l(\langle \mathbf{w}, \mathbf{x}^{(i)} \rangle, \text{ y}^{(i)}).$$

In Proposition 1 we show a bound on the bias $b_d$ incurred by only using features in $\mathcal{S}_d$ to fit the target predictor, as opposed to using all possible $d$-dimensional linear features. Note that the upper bound on $b_d$ reduces to 0 as we add more features $d \to D$. The rate at which $b_d \to 0$ is determined by how the optimal linear classifier $\boldsymbol{w}_T^\star$ (on target) interplays with the projection matrix $\mathsf{M}_d$ learned on the source data. On one extreme, when there is no distribution shift, we know that for the projection $\Pi_1$ returned by PRO$^2$, $\Pi_1 \propto \boldsymbol{w}_T^\star$, and thus $(\boldsymbol{I}_D - \mathsf{M}_1)\boldsymbol{w}_T^\star = 0$, the bias $b_d \to 0$ with just one direction. On the other extreme, if $\Pi_d$ consists of random orthogonal features, then the bias $b_d$ decreases at a

Figure 3: **Evaluation of $\text{PRO}^2$ on shifted homoscedastic Gaussian data.** *(left)* The x- and y-axes denote the rank of $\mathsf{M}_d$ and the nullspace norm $\|(\boldsymbol{I}_D - \mathsf{M}_d)\boldsymbol{w}_T^\star\|_2$, respectively. The nullspace norm drops slowly for more severe distribution shifts, indicating that $b_d$ would be low enough only at very high values of $d$ for severe shifts. *(right)* For less severe distribution shifts (ID and Near OOD), low-dimensional projections suffer from less bias, resulting in higher accuracy in the low-data regime. For the Far OOD distribution, using all 20-dimensional features is best, as bias drops more slowly.

rate $\mathcal{O}(\sqrt{D-d})$ even when there is distribution shift. This indicates that the rate at which the bias reduces as we increase $d$ should be controlled by degree of distribution shift, and how informative the source features (in $\Pi_d$) remain under this shift. This argument is captured more formally in the following Proposition.

**Proposition 1** (bias induced by distribution shift)**.** *For some $\boldsymbol{w}_T^\star$ that is the optimal linear predictor on distribution $p_T$ over the full feature space, and an $L$-Lipschitz smooth convex loss $l$, the bias $b_d \leq L \cdot \|(\boldsymbol{I}_D - \mathsf{M}_d)\boldsymbol{w}_T^\star\|_2$. When $\mathsf{M}_d$ is a random rank $d$ projection matrix with columns drawn uniformly over the sphere $S^{D-1}$, then $b_d L \sqrt{D-d} \cdot \|\boldsymbol{w}_T^\star\|_2$.*

In Theorem 2, we describe the full bias-variance tradeoff where we see that the variance term is also controlled by the number of features $d$ but unlike the bias is independent of the nature of shift between source and the target.

**Theorem 2** (bias-variance tradeoff)**.** *When the conditions in Lemma 1 hold and when $\|\mathbf{x}\|_\infty = \mathcal{O}(1)$, for $B$-bounded loss $l$, w.h.p. $1 - \delta$, the excess risk for the solution $\hat{\mathbf{w}}_d$ of $\text{PRO}^2$ that uses $d$ features is $\mathcal{L}_T(\hat{\mathbf{w}}_d) - \min_{\mathbf{w} \in \mathcal{W}} \mathcal{L}_T(\mathbf{w})$*

$$\|(\boldsymbol{I}_D - \mathsf{M}_d)\boldsymbol{w}_T^\star\|_2 + \frac{\sqrt{d} + B\sqrt{\log(1/\delta)}}{\sqrt{M}}, \tag{2}$$

*where the first term controls the bias and the second controls the variance.*

This result provides insights on what factors affect generalization when probing on target data. Tighter compression of the original representation, i.e., using a smaller $d$, increases bias while decreasing variance. The rate of bias increase is determined by the degree of distribution shift, where more severe shifts correspond to a steeper increase in bias. However, this bias can be mitigated as long as the important features needed for prediction on the target domain are covered by the compressed representation. Thus, $\text{PRO}^2$ induces a favorable bias-variance tradeoff, as the features extracted are predictive and diverse and hence are more likely to cover the important features needed for the target domain, allowing compression to a smaller $d$ while still maintaining low bias. The distribution shift has no effect on variance, and variance can only be decreased by using a low-dimensional represent (at the cost of bias) or learning from a larger target dataset (higher $M$).

## 5.2 BIAS-VARIANCE TRADEOFF IN A SIMPLIFIED GAUSSIAN MODEL.

In this subsection, we consider a simplified setting of a shifted homoscedastic Gaussian (SHOG). Within this model, we show that the more general statement in Theorem 2 can be simplified further to provide a more intuitive relationship between the factors that affect excess risk on target. We also empirically demonstrate the behavior predicted by our bounds on synthetic SHOG data.

**Shifted homoscedastic Gaussian (SHOG) model of distribution shift.** We model the source distribution as a Bernoulli mixture model of data in which binary labels are balanced ($\text{y} \sim \text{Bern}(0.5)$) and the class conditional distributions are homoscedastic multivariate Gaussians:

$$\mathbf{x} \mid \text{y} \sim \mathcal{N}(\mu_{\text{y}}, \Sigma_S) \quad \text{for} \quad \text{y} \in \{0, 1\},$$

where $\mu_0, \mu_1 \in^D$ are mean vectors and $\Sigma_S \in^{D \times D}$ is the shared covariance matrix. The target distribution has the same label distribution and Gaussian means, but a different covariance matrix given

by $\Sigma_T$. Under this model, the degree of shift is fully realized by the distance between $\Sigma_S, \Sigma_T$ under some reasonable metric. This directly impacts the bias term $b_d$ when $\Pi_d$ is either returned by PRO$^2$ or a random projection matrix with columns drawn uniformly over the sphere $S^{d-1}$. We specialize the more general bias-variance tradeoff result to a shifted homoscedastic Gaussian (SHOG) model in Corollary 3, where we derive a simpler bound distilling the effect of $d$, and the degree of distribution shift on the target excess risk.

**Corollary 3** (tradeoff under SHOG). *Under SHOG model, and conditions for random projection* $\mathsf{M}_d$ *in Proposition 1, the target excess risk* $\mathcal{L}_T(\hat{\mathbf{w}}_d) - \mathcal{L}_T(\boldsymbol{w}_T^\star)\sqrt{D-d} \cdot \mathrm{KL}(p_S||p_T) + \sqrt{\frac{d}{M}}$.

Recall from Proposition 1, the bias scales directly with the norm along the null space of $\mathsf{M}_d$. In Fig. 3*(left)*, we plot the nullspace norm for different $d$ in three target distributions of varying distribution shift severity under the SHOG model. We see that the more severe shifts have a higher nullspace norm even at large values of $d$. This indicates that the excess risk on target (OOD) may suffer from high bias on severe shifts. In Fig. 3*(right)*, we see that the source distribution (ID) suffers from virtually no bias, since $d{=}1$ achieves optimal target accuracy for even small target datasets. In contrast, the "Near OOD" and "Far OOD" distributions suffer from high bias since even with large datasets they fall short of optimal target performance (by $>40\%$), and higher projection dimension $d$ is needed for adaptation. This also validates our findings in Theorem 2 and Corollary 3.

## 6 EXPERIMENTS

In this section, we aim to empirically answer the following questions: (1) Can PRO$^2$ identify a feature-space basis for rapid adaptation, and how does it compare to other methods for extracting features? (2) How does the dimensionality of the feature-space basis affect sample efficiency in different distribution shift conditions? We provide additional empirical results and analyses, such as showing that the adaptation performance of PRO$^2$ improves with better pre-trained backbones, in Appendix C. Details on pre-trained models and training details are in Appendix B.

### 6.1 EXPERIMENTAL SETUP

**Datasets.** We run experiments on six datasets with distribution shifts: 4-way collages (Teney et al., 2021), Waterbirds (Sagawa et al., 2020), CelebA (Liu et al., 2015), Camelyon (Bandi et al., 2018), Living17 (Santurkar et al., 2020), and FMoW (Koh et al., 2021) datasets. Each of these datasets have a source distribution that we use for training. For the first four datasets, we construct multiple target distributions for evaluation, representative of a range of potential test distributions. For the latter two datasets, which are larger datasets representing shifts that may occur in the wild, we evaluate on the given test set. For all settings, we use the original source datasets, which each contain thousands of datapoints. For target data, we subsample very small label-balanced datasets for adaptation, with $\{2, 8, 32, 128\}$ images per label for the first four datasets and $\{1, 2, 5\}$ images per label for the latter two datasets. The remaining target distribution datapoints are used for evaluation. Due to space constraints, we describe the different target distributions in Appendix B.

**Computational efficiency.** Similarly to Mehta et al. (2022), we use feature embeddings from a pre-trained backbone without fine-tuning. Our aim is to develop methods that can leverage pretrained models out-of-the-box with minimal computational requirements: our training involves at most two linear layers on top of cached feature vectors. For all comparisons, we hyperparameter tune over 3 different learning rates (0.1, 0.01, and 0.001) as well as 3 different $L2$ regularization weights (0.1, 0.01, 0.001). In our main experiments in Sec. 6.2, we also sweep over 6 different projection dimensions ($d = 1, 4, 16, 64, 256, 1024$) and report results over 10 runs. For hyperparameter tuning, we adopt the typical practice of using a target validation set, which is common in prior work in similar transfer learning settings (Kirichenko et al., 2022; Mehta et al., 2022; Lee et al., 2022a). The challenge of hyperparameter tuning for a target domain without additional domain-specific information remains an open problem that we hope can be addressed in future work. As a demonstration of the computational efficiency of PRO$^2$, after caching pre-trained embeddings, we can collectively run all experiments in Sec. 6.2, which is nearly 30k runs due to hyperparameter tuning, within 24 *hours* using four standard CPUs and *no GPUs*. We find that PRO$^2$ is robust to learning rate, which is expected as the optimization problem is linear.

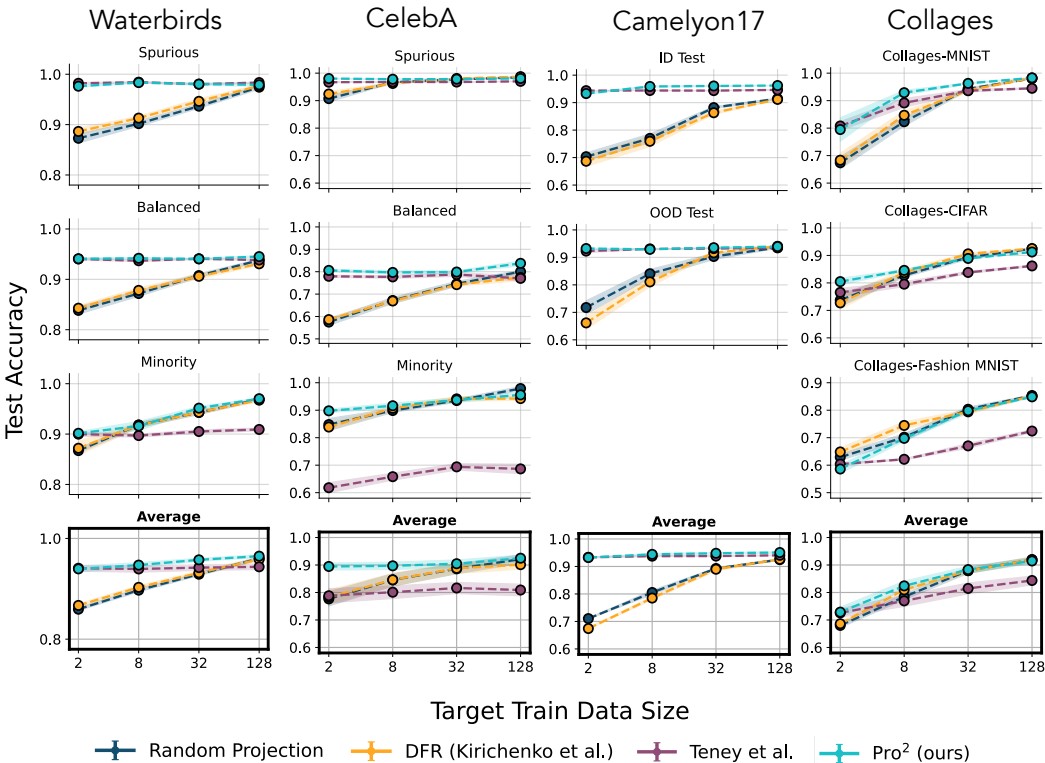

Figure 4: **Main results.** We compare 4 different methods for learning features to adapt to a target distribution: (1) Random Projection, (2) DFR Kirichenko et al. (2022), (3) Teney et al. (2021), and (4) PRO². We report average target accuracies after probing with different target dataset sizes ranging from 2 to 128 datapoints per label; error bars indicate standard error across 10 runs. PRO² is the best performing or tied for best performing method *across each of these 4 datasets with any amount of target data*. PRO² substantially outperforms Random Projection and DFR in the low-data regime on all four datasets. PRO² also outperforms Teney et al. (2021) on average on 3 of the 4 datasets particularly when given more target data.

| | Living17 | | | FMoW | | |
|---|---|---|---|---|---|---|
| Target Train Data Size (per label) | 1 | 2 | 5 | 1 | 2 | 5 |
| Random Projection | 85.7 (0.6) | 92.7 (1.0) | **99.2 (0.1)** | 16.3 (0.7) | 23.6 (0.6) | 33.3 (0.6) |
| DFR (Kirichenko et al.) | 87.1 (0.9) | 95.0 (0.9) | 98.8 (0.3) | 17.5 (0.8) | 24.0 (0.6) | 35.1 (0.6) |
| PRO² | **91.2 (0.4)** | **95.7 (0.7)** | **99.2 (0.06)** | **20.2 (0.9)** | **28.7 (1.1)** | **37.2 (0.8)** |

Table 1: **Additional main results.** We run additional experiments on the Living17 dataset from the Breeds benchmark (Santurkar et al., 2020) and FMoW (Koh et al., 2021), reporting adaptation accuracy and standard error across 10 runs. Both of these datasets are challenging multi-class distribution shift tasks and are representative of real-world scenarios. We find that similar to the other datasets, PRO² is the best performing or tied for best performing method on these datasets when given a limited amount of target data.

## 6.2    COMPARISON TO PRIOR PROJECTION METHODS

We investigate whether PRO² can extract features that can facilitate adaptation to different distribution shifts, and how it compares other feature extraction methods. We perform a comprehensive experimental evaluation on the six datasets, comparing PRO² against four other projection methods: (1) Random Projection, (2) DFR Kirichenko et al. (2022), which uses standard linear probing, and (3) Teney et al. (2021), which aims to learn multiple predictive patterns by minimizing the alignment of input gradients over pairs of features. Experiments in Fig. 4 and Tab. 1 indicate that across all different target distributions six datasets, PRO² significantly outperforms Random Projection and DFR, especially in the low-data regime. In particular, these results show that DFR or standard linear probing, the strategy adopted by several additional prior works by default Mehta et al. (2022); Izmailov et al. (2022), is not the most data-efficient way to utilize pre-trained embeddings when given limited target data. This is because such embeddings contain redundant or non-predictive information, and including these features during adaptation leads to higher variance without decreasing bias, which in turn means that we need more labeled samples. In contrast, PRO² improves sample efficiency

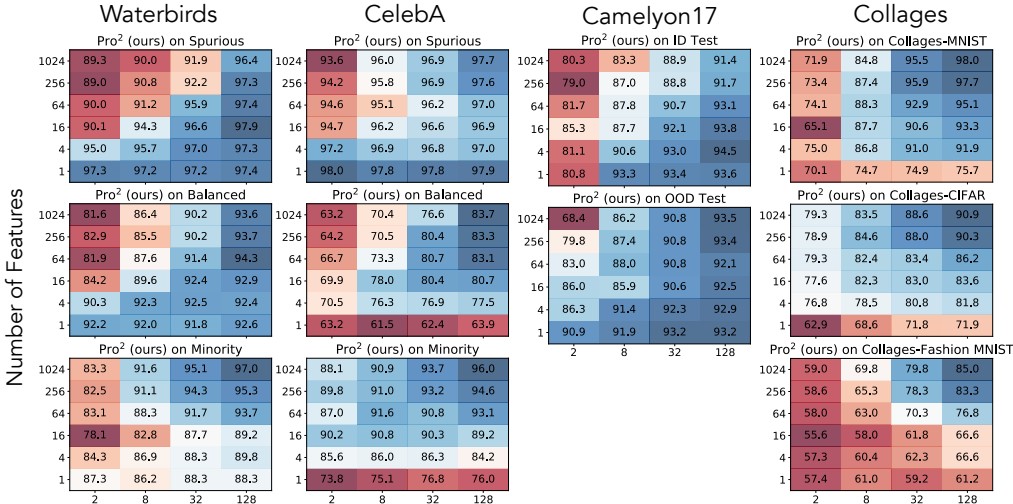

Figure 5: **Feature-space dimensionality of PRO$^2$ and severity of distribution shift.** We vary the feature-space dimensions $d$ (y-axis) of PRO$^2$ and report held-out accuracy after training on target datasets of different size (x-axis) on our 4 datasets. Higher accuracies are in blue and lower accuracies are in red. We see that smaller feature-space dimensions suffice for target distributions with milder distribution shifts while higher dimensions are required for more severe shifts. For example, on the spurious test distribution (small dist. shift) of Waterbirds/CelebA, the bottom row, which uses $d = 1$ is bluest, while the blue is concentrated in the top right squares (which use more features and more data) for more difficult distribution shifts such as Minority for Waterbirds/CelebA and the collages test sets.

by first extracting a predictive feature-space basis from the source distribution, removing redundant information. Teney et al. (2021) is sufficient in some scenarios with milder distribution shift, where a diverse range of features are not needed for adaptation. However, it fails to achieve high accuracy given a large target dataset on more severe distribution shifts, such as the Minority distributions on Waterbirds and CelebA or the Fashion-MNIST and CIFAR distributions in 4-Way Collages. This indicates that the feature diversity from the orthogonality constraint gives PRO$^2$ better coverage of different features, enabling better adaptation to severe distribution shifts given enough target data. These results demonstrate the effectiveness of PRO$^2$ compared to existing methods in the few-shot adaptation problem setting.

## 6.3 PROJECTION DIMENSION AND SHIFT SEVERITY

In this subsection, we investigate how the feature-space dimension $d$ affects the sample efficiency of PRO$^2$, for different degrees of distribution shift. Experiments in Fig. 5 show that when the distribution shift is less severe, such as the Spurious test distributions on Waterbirds and CelebA, it is helpful to reduce the number of features used. This scenario is analogous to the ID setting in Fig. 3. In such scenarios, the top-ranked features from the source data are also predictive on the target distribution, and incorporating additional features worsens generalization because it increases variance without sufficiently decreasing bias. However, when the distribution shift is more severe, such as the Minority distributions on Waterbirds and CelebA or Collages-Fashion MNIST and Collages-CIFAR, it is helpful to increase the number of features used. This scenario is analogous to the Far OOD setting in Fig. 3. These empirical results are supported formally by our theoretical results in Sec. 5, which show that the optimal number of features to use increases with distribution shift severity.

## 7 CONCLUSION

In this paper, we propose PRO$^2$, a lightweight framework consisting of 2 steps: (1) a projection step that extracts a diverse and predictive feature-space basis and (2) a probing step that interpolates between the projected features to efficiently adapt varying target distributions. Our theoretical and empirical analyses reveal a number of interesting novel insights: (i) standard linear probing is not the best approach for few-shot adaptation; (ii) Retaining a diverse range of potentially useful features that different target distributions may require improves sample efficiency, (iii) we can trade off how much to adapt (size of the feature-space basis) vs number of samples, picking the best basis to adapt

for each level of shift. These insights open up a range of exciting paths for future work. First, our framework may be extended to other problem settings, such as the active learning setting, in which the model can adaptively request target labels. Another interesting direction would be selecting which features to use in an unsupervised fashion, without any labeled target data. For limitations, more complex target distributions generally require more directions and therefore more data to fit. Our analysis on the bias-variance tradeoff expressed in Theorem 2 depends on the dataset size, so $\text{PRO}^2$ will have smaller benefits in settings with a large amount of target data.

## ACKNOWLEDGMENTS

We thank members of the IRIS and RAIL labs for helpful discussions on this project. This work was supported by NSF, KFAS, Apple, Juniper, and ONR grant N00014-20-1-2675.

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

# A  PROOFS FOR THEORETICAL ANALYSIS

We present proofs for our theoretical analysis in Sec. 5 along with some additional statements. As in the main paper, we denote the dimensionality of the feature-space basis learned by $\text{PRO}^2$ as $d$, the original dimension of the representations given by the feature backbone $f$ as $D$, source and target distributions as $p_S$ and $p_T$, and the number of source and target datapoints as $N$ and $M$. We let $\mathsf{M}_d$ denote the projection matrix for $\text{span}(\{\Pi_i\}_{i=1}^d)$, $\mathsf{M}_d = [\Pi_1, .., \Pi_d][\Pi_1, .., \Pi_d]^\top$. If the target error for the feature $w$ is $\mathcal{L}_T(\mathbf{w}) := \mathbb{E}_{\mathcal{D}_T} l(\langle \mathbf{w}, \mathbf{x} \rangle, \mathbf{y})$, then the bias incurred by probing on the subspace $\mathsf{M}_d$ consisting of source features is:

$$b_d := \min_{\mathbf{w}' \in \text{span}(\{\Pi_i\}_{i=1}^d)} \mathcal{L}_T(\mathbf{w}') - \min_{\mathbf{w} \in \mathcal{W}} \mathcal{L}_T(\mathbf{w}),$$

and we denote the feature-space basis of dimensionality $d$ learned by $\text{PRO}^2$ as follows:

$$\hat{\mathbf{w}}_d := \underset{\mathbf{w} \in \text{span}(\{\Pi_i\}_{i=1}^d)}{\arg\min} \sum_{i=1}^M l(\langle \mathbf{w}, \mathbf{x}^{(i)} \rangle, \mathbf{y}^{(i)}). \tag{3}$$

From the original $D$-dimensional feature representations given by our feature backbone $f$, we want our learned linear projections $\Pi :^D \to^d$ to retain as much information as possible that is relevant in predicting the label y. In other words, we want to maximize the mutual information between the projected features $\Pi(\mathbf{x})$ and the labels y. In Theorem 6, We first formally characterize the solution found by the projection step in $\text{PRO}^2$ as maximizing this mutual information amongst all rank $d$ matrices with orthogonal columns, using the following two lemmas.

**Lemma 4** (Entropy of a sub-gaussian under low rank projection). *For a $\sigma-$sub-gaussian random variable $\mathbf{x}$, and rank $r$ orthonormal matrix $\boldsymbol{A} \in^{d \times D}$ the differential entropy $H(\boldsymbol{A}\mathbf{x})$ for projection $\boldsymbol{A}\mathbf{x}$ is $\mathcal{O}(1/d)$ when $\log \sigma = \mathcal{O}(1/d^2)$.*

*Proof.* Let $\boldsymbol{A}_i$ denote the $i^{th}$ row of $\boldsymbol{A}$, then:

$$H(\boldsymbol{A}\mathbf{x}) = H([\boldsymbol{A}_1^\top \mathbf{x}, \ldots, \boldsymbol{A}_r^\top \mathbf{x}]^\top) = H(\boldsymbol{A}_1^\top \mathbf{x}) + \sum_{i=2}^{i=d} H(\boldsymbol{A}_i^\top \mathbf{x} \mid \boldsymbol{A}_1^\top \mathbf{x}, \ldots, \boldsymbol{A}_{i-1}^\top \mathbf{x}) \le \sum_{i=1}^{i=d} H(\boldsymbol{A}_i^\top \mathbf{x})$$

Since, $\mathbf{x}$ is $\sigma-$sub-gaussian, using standard bounds on differential entropy and the inequality above we can argue that:

$$\mathbb{E} e^{t^2 \boldsymbol{A}_i^\top \mathbf{x}} \le e^{t^2 \sigma^2 / 2}, \quad \forall i, t$$

$$\implies H(\boldsymbol{A}_i^\top \mathbf{x}) \le \frac{1}{2} \log(2\pi e \sigma^2) \ \forall i$$

$$\implies H(\boldsymbol{A}_i^\top \mathbf{x})(1/d^2) \ \forall i \ (\text{since } \log \sigma = \mathcal{O}(1/d^2))$$

$$\implies H(\boldsymbol{A}\mathbf{x}) = \mathcal{O}(1/d)$$

$\square$

**Lemma 5** (Entropy for a mixture of $\sigma-$sub-gaussians). *For a $d-$dimensional random variable $\mathbf{x}$ that is a mixture of two $\sigma-$sub-gaussian random variables $\mathbf{x}_1$ and $\mathbf{x}_2$ with overlapping supports, bounded Jensen-Shannon divergence $\text{JS}(\boldsymbol{v}^\top \mathbf{x}_1 \| \boldsymbol{v}^\top \mathbf{x}_2) \le \beta$ and mixture proportion $\alpha \in [0, 1]$ the density function $p(\mathbf{x}) = \alpha \cdot p_1(\mathbf{x}) + (1 - \alpha) \cdot p_2(\mathbf{x})$, then the entropy $H(\boldsymbol{v}^\top \mathbf{x})$ for $\|\boldsymbol{v}\|_2 = 1$, is at most $\mathcal{O}(\log \sigma + \beta)$.*

*Proof.* Using Jensen inequality and the definition of KL divergence, the differential entropy can be broken down as follows:

$$H(\boldsymbol{v}^\top \mathbf{x}) = -\int (\alpha \cdot p_1(\boldsymbol{v}^\top \boldsymbol{x}) + (1 - \alpha) \cdot p_2(\boldsymbol{v}^\top \boldsymbol{x})) \log(\alpha \cdot p_1(\boldsymbol{v}^\top \boldsymbol{x}) + (1 - \alpha) \cdot p_2(\boldsymbol{v}^\top \boldsymbol{x})) \, d\boldsymbol{x}$$

$$\le \alpha^2 H(\boldsymbol{v}^\top \mathbf{x}_1) + (1 - \alpha)^2 H(\boldsymbol{v}^\top \mathbf{x}_2)$$

$$- \alpha(1 - \alpha) \int p_1 \boldsymbol{v}^\top \boldsymbol{x}) \log p_2(\boldsymbol{v}^\top \boldsymbol{x}) \, d\boldsymbol{x} - \alpha(1 - \alpha) \int p_2(\boldsymbol{v}^\top \boldsymbol{x}) \log p_1(\boldsymbol{v}^\top \boldsymbol{x}) \, d\boldsymbol{x}$$

$$\le \alpha H(\boldsymbol{v}^\top \mathbf{x}_1) + (1 - \alpha) H(\boldsymbol{v}^\top \mathbf{x}_2) + 2\alpha(1 - \alpha)\beta$$

$$\log(\sigma) + \beta$$

where the last step follows from the first two arguments made in the proof for Lemma 4. $\square$

**Theorem 6** (Information in projected input). *When the distributions $p(\mathbf{x} \mid \mathbf{y})$ are $\exp(d^{-2})-sub$-gaussian for each $y$ and the Jensen-Shannon divergence $\mathrm{JS}(p(\boldsymbol{v}^\top \mathbf{x} \mid y = 0) \parallel p(\boldsymbol{v}^\top \mathbf{x} \mid y = 1)) = \mathcal{O}(1/d)$. the solution $\{\Pi_i\}_{i=1}^d$ returned by $\mathrm{PRO}^2$ maximizes a tight lower bound (difference bounded by an $O(1)$ constant) on the mutual information criterion $I(\mathbf{Ax}; \mathbf{y})$ among all $d \times D$ row-orthonormal matrices $\mathbf{A}$. (See end of the proof for **discussion on assumptions** and **remark on tightness**).*

*Proof.* We use an inductive argument on $d$. Consider the following maximization problem where $\mathbb{B}_d$ is the set of all row orthonormal matrices of row rank $d \ll D$:

$$\max_{\boldsymbol{A} \in \mathbb{B}_d} I(\boldsymbol{A}\mathbf{x}; \mathbf{y}). \tag{4}$$

Let $d > 1$. Then, we can re-write the above as:

$$\max_{\boldsymbol{A} \in \mathbb{B}^{d \times D}} I(\boldsymbol{A}\mathbf{x}; \mathbf{y}) = \max_{\boldsymbol{A}' \in \mathbb{B}_{d-1}, \boldsymbol{v} \in D} I\left(\boldsymbol{A}'\mathbf{x}, \boldsymbol{v}^\top \mathbf{x}^\top; \mathbf{y}\right) \quad \text{where,} \quad \boldsymbol{v} \in \mathrm{NullSpace}(\boldsymbol{A}'), \|\boldsymbol{v}\|_2 = 1. \tag{5}$$

Now, we can decompose this expression using the conditional mutual information identities:

$$I\left(\boldsymbol{A}'\mathbf{x}, \boldsymbol{v}^\top \mathbf{x}^\top; \mathbf{y}\right) = I(\boldsymbol{A}'x; \mathbf{y}) + I(\boldsymbol{v}^\top \mathbf{x}; \mathbf{y}) - I(\boldsymbol{v}^\top \mathbf{x}; \boldsymbol{A}'\mathbf{x}) + I(\boldsymbol{v}^\top \mathbf{x}; \boldsymbol{A}'\mathbf{x} \mid y)$$

$$= I(\boldsymbol{A}'x; \mathbf{y}) + I(\boldsymbol{v}^\top \mathbf{x}; \mathbf{y}) - \left(I(\boldsymbol{v}^\top \mathbf{x}; \boldsymbol{A}'\mathbf{x}) - I(\boldsymbol{v}^\top \mathbf{x}; \boldsymbol{A}'\mathbf{x} \mid y)\right) \tag{6}$$

Now, we upper bound the drop in information when we condition on y: $(I(\boldsymbol{v}^\top \mathbf{x}; \boldsymbol{A}'\mathbf{x}) - I(\boldsymbol{v}^\top \mathbf{x}; \boldsymbol{A}'\mathbf{x} \mid y))$ using Lemma 4 and Lemma 5.

$$I(\boldsymbol{v}^\top \mathbf{x}; \boldsymbol{A}'\mathbf{x}) - I(\boldsymbol{v}^\top \mathbf{x}; \boldsymbol{A}'\mathbf{x} \mid y) = H([\boldsymbol{v}^\top \mathbf{x}; \boldsymbol{A}'\mathbf{x}]^\top \mid \mathbf{y}) - H(\boldsymbol{v}^\top \mathbf{x} \mid \mathbf{y}) - H(\boldsymbol{A}'\mathbf{x} \mid \mathbf{y}) + I(\boldsymbol{v}^\top \mathbf{x}; \boldsymbol{A}'\mathbf{x})$$

$$\leq H([\boldsymbol{v}^\top \mathbf{x}; \boldsymbol{A}'\mathbf{x}]^\top \mid \mathbf{y}) + H(\boldsymbol{v}^\top \mathbf{x}) = H(\boldsymbol{A}\mathbf{x} \mid \mathbf{y}) + H(\boldsymbol{v}^\top \mathbf{x})$$

$$\log \sigma + \beta = \mathcal{O}(1/d), \tag{7}$$

where the last statement applies Lemma 4 to bound $H(\boldsymbol{A}\mathbf{x} \mid \mathbf{y})$ (since $\boldsymbol{A}\mathbf{x} \mid y = 0$ and $\boldsymbol{A}\mathbf{x} \mid y = 1$ are $\sigma-$sub-gaussian) and Lemma 5 to bound the entropy on the mixture of sub-gaussians $H(\boldsymbol{v}^\top \mathbf{x})$. Also, note that the conditional distributions differ in Jensen-Shannon divergence by $\mathcal{O}(1/d)$ and the sub-gaussian assumption gives us $\log \sigma = \mathcal{O}(1/d^2)$.

Using equation 6, equation 7 we have:

$$\max_{\boldsymbol{A} \in \mathbb{B}_d} I(\boldsymbol{A}\mathbf{x}; \mathbf{y}) \quad \geq \quad \max_{\substack{\boldsymbol{A}' \in \mathbb{B}_{d-1}, \\ \boldsymbol{v} \in \mathrm{NullSpace}(\mathrm{A}), \|\boldsymbol{v}\|_2 = 1}} I\left(\boldsymbol{A}'\mathbf{x}; y\right) + I(\boldsymbol{v}^\top \mathbf{x}; \mathbf{y}) - \mathcal{O}(1/d). \tag{8}$$

Let $\boldsymbol{A}_i$ denote the $i^{th}$ row of $\boldsymbol{A}$, then. Then, applying the above inductively for all $d$:

$$\max_{\boldsymbol{A} \in \mathbb{B}_d} I(\boldsymbol{A}\mathbf{x}; \mathbf{y}) \quad \geq \quad \max_{\boldsymbol{A} \in \mathbb{B}_d} \left(\sum_{i=1}^{i=d} I(\boldsymbol{A}_i^\top \mathbf{x}; y)\right) - \mathcal{O}(1) \tag{9}$$

Let $\boldsymbol{A}^*$ be the solution of the right hand side and $\boldsymbol{v}^* = \arg\max_{\boldsymbol{v}: \|\boldsymbol{v}\|_2 = 1} I(\boldsymbol{v}^\top \mathbf{x}; \mathbf{y})$. Next, we note that $\exists i$ such that $\boldsymbol{A}_i^* = \boldsymbol{v}^*$. It is easy to prove this by contradiction. Consider the case where $\nexists i$ such that $\boldsymbol{A}_i^* = \boldsymbol{v}^*$. Then, we can construct a solution $\{(\mathbf{I}_D - \boldsymbol{v}^* \boldsymbol{v}^{*\top}) \boldsymbol{A}_i^*\}_{i=1}^d$, order them by mutual information $I((\boldsymbol{A}_i^*)^\top (\mathbf{I}_D - \boldsymbol{v}^* \boldsymbol{v}^{*\top})\mathbf{x}; \mathbf{y})$, take the top $d-1$ entries and append to this set $\boldsymbol{v}^*$. The new solution would have a higher value of the objective on the right hand side of equation 9, since the new solution retains optimal directions perpendicular to $\boldsymbol{v}^*$ while adding $\boldsymbol{v}^*$ to the set. Thus, we arrive at a contradiction and it is clear that $\boldsymbol{v}^*$ belongs to the solution $\boldsymbol{A}^*$ for the objective on the right side of equation 9.

Knowing that $\boldsymbol{v}^*$ has to be part of $\boldsymbol{A}*$, we can now write the right side of equation 9 as the following:

$$\max_{\boldsymbol{A} \in \mathbb{R}^{d \times D}} I(\boldsymbol{A}\mathbf{x}; \mathbf{y}) \geq \max_{\boldsymbol{v}_1 \in \mathbb{R}^D} I\left(\boldsymbol{v}_1^\top \mathbf{x}; \mathbf{y}\right)$$

$$+ \max_{\boldsymbol{v}_2 \in \mathbb{R}^D} I\left(\boldsymbol{v}_2^\top \left(I - \boldsymbol{v}_1^* \boldsymbol{v}_1^{*\top}\right) \mathbf{x}; \mathbf{y}\right)$$

$$+ \max_{\boldsymbol{v}_3 \in \mathbb{R}^D} I\left(\boldsymbol{v}_3^\top \left(I - \boldsymbol{v}_2^* \boldsymbol{v}_2^{\star\top}\right) \left(I - \boldsymbol{v}_1^\star \boldsymbol{v}_1^{*\top}\right) \mathbf{x}; \mathbf{y}\right) + \ldots - \mathcal{O}(1), \tag{10}$$

where $\boldsymbol{v}_1^\star, \boldsymbol{v}_2^\star, \ldots, \boldsymbol{v}_d^\star$ denote the solutions to each subsequent max term. This sequence of solutions is the same as that returned by solving the following iterative optimization problem because maximizing mutual information with label for a linear projection of the input is the same as finding a direction that minimizes Bayes error of the linear projection (Petridis and Perantonis (2004) connects mutual information to cross entropy loss and Bayes error):

1. $\boldsymbol{v}_1^* = \arg\min_{\|\boldsymbol{v}\| \leq 1} l(\langle \boldsymbol{v}, \mathbf{x} \rangle, \mathbf{y})$

2. Project data in the null space of $\boldsymbol{v}_1^*$: $(I - \boldsymbol{v}_1^* \boldsymbol{v}_1^{*\top})\mathbf{x}$

3. Re-solve (1.) to get next $\boldsymbol{v}_i^*$ and so on.

Finally, it is easy to see that solution returned by the above iterative optimization is the same as that returned by the project step of $\textsc{Pro}^2$.

**Discussion on assumptions:** Following are some remarks and intuitions behind the assumptions we make:

- **Sub-gaussianity**: We need sub-gaussianity to bound the entropy of linear projections, which is easily satisfied for inputs with bounded support. Note that the sub-gaussian parameter $\sigma$ need only satisfy $\log \sigma = \mathcal{O}(1/d^2)$, where $d \ll D$ which is the input dimension of the data.

- **Bounded JS-divergence**: The main intuition behind why we need the class conditional distributions to not differ too much (bounded JS-divergence) along linear projections is that if they are very different from each other it is possible that even with sub-gaussianity assumptions there may exist linear projections that have a high mutual information over the mixture of conditionals (which is the marginal input distribution $p(\mathbf{x})$ i.e., $I(\boldsymbol{v}^\top \mathbf{x}; \boldsymbol{A}'\mathbf{x})$ is high) but not when we condition on the label (i.e., $I(\boldsymbol{v}^\top \mathbf{x}; \boldsymbol{A}'\mathbf{x} \mid \mathbf{y})$ is low). Now, since we iteratively search for linear projections, our project step is oblivious to these interactions and we may recover both of these directions (see equation 6 and Lemma 5). But only one may be present in the information theoretically optimal linear projection.

**Remark on tightness of our lower bound:** We show that we maximize a lower bound in equation 9. But, in the special case when the class conditionals are log-concave (e.g., multivariate Gaussian) we can also show something much tighter: $\max_{\boldsymbol{A} \in \mathbb{B}_d} I(\boldsymbol{A}\mathbf{x}; \mathbf{y}) = \max_{\boldsymbol{A} \in \mathbb{B}_d} \left( \sum_{i=1}^{i=d} I(\boldsymbol{A}_i^\top \mathbf{x}; y) \right) - \Theta(1)$. This is because our upper bounds on the entropy terms have matching lower bounds for log-concave distributions, which can then be applied to lower bound the negative terms in the first step of equation 6.

$\square$

We now provide proofs of the generalization bounds in Section 5 showing the bias-variance tradeoff.

**Lemma 7** (generalization bound for probing projected features). *For an $L$-Lipshitz, $B$-bounded loss $l$, with probability $\geq 1 - \delta$, $\hat{\mathbf{w}}_d$ in equation 3 has generalization error $\frac{\sqrt{d} + B\sqrt{\log(1/\delta)}}{\sqrt{M}}$, when $\|\mathbf{x}\|_\infty = O(1)$.*

*Proof.* For this proof, we invoke the following two lemmas.

**Lemma 1** (generalization bound for linear functions Bartlett and Mendelson (2002)). *For an $L$-Lipshitz $B$-bounded loss $l$, the generalization error for predictor $\hat{\mathbf{w}}_d$, contained in the class of $l_2$ norm bounded linear predictors $\mathcal{W}$ is bounded with probability $\geq 1 - \delta$:*

$$l(\langle \hat{\mathbf{w}}_d, \mathbf{x} \rangle, \mathbf{y}) - \sum_{i=1}^{M} l(\langle \mathbf{w}, \mathsf{M}_d \mathbf{x}^{(i)} \rangle, \mathbf{y}^{(i)}) \leq 2L\mathcal{R}_n(\mathcal{W}) + B\sqrt{\frac{\log(1/\delta)}{2M}}$$

*where $\mathcal{R}_n(\mathcal{W})$ is the empirical Rademacher complexity of $l_2$ norm bounded linear predictors.*

**Lemma 2** ($\mathcal{R}_n(\mathcal{W})$ bound for linear functions Kakade et al. (2008)). *Let $\mathcal{W}$ be a convex set inducing the set of linear functions $\mathcal{F}(\mathcal{W}) \triangleq \{\langle \mathbf{w}, \mathbf{x} \rangle : \mathcal{X} \mapsto | \ w \in \mathcal{W}\}$ for some input space $\mathcal{X}$, bounded in norm $\|\cdot\|$ by some value $R > 0$. If there exists a mapping $h : \mathcal{W} \mapsto$ that is $\kappa$-strongly convex with respect to the dual norm $\|\cdot\|_*$ and some subset $\mathcal{W}' \subseteq \mathcal{W}$ takes bounded values of $h(\cdot)$ $\{h(\mathbf{w}) \leq K \ | \ \mathbf{w} \in \mathcal{W}'\}$ for some $K > 0$, then the empirical Rademacher complexity of the subset $\mathcal{W}'$ is bounded by $\mathcal{R}_n(\mathcal{F}(\mathcal{W}')) \leq R\sqrt{\frac{2K}{\kappa n}}$.*

Let $\|\cdot\|_2^2$ be the function $h : \mathcal{W} \mapsto$ in Lemma 2; we know that $\|\cdot\|_2^2$ is 2-strongly convex in $l_2$ norm. Further, take the standard $l_2$ norm as the norm over $\mathcal{X}$. So, the dual norm $\|\cdot\|_*$ is also given by $l_2$ norm. Thus, $\kappa = 2$. We also know that $\mathcal{W}$ is bounded in $\|\cdot\|_2$ by 1, based on our setup definition. Thus, $K = 1$.

Further, we note that $\|\mathbf{x}\|_\infty = O(1)$. We apply Cauchy-Schwartz and use the fact that $\mathsf{M}_d = 1$ to bound the norm of the projected vector:

$$\|\mathsf{M}_d \mathbf{x}\| \leq \mathsf{M}_d \|\mathbf{x}\|_2 \leq \mathsf{M}_d \sqrt{d} \|\mathbf{x}\|_\infty \sqrt{d}. \tag{11}$$

By Lemma 2 we get the empirical Rademacher complexity $\mathcal{R}_M(\mathcal{W})\sqrt{d/M}$, and plugging this into Lemma 1 yields the main result in Lemma 7.

$\square$

**Theorem 8** (bias-variance tradeoff, Theorem 2). *When the conditions in Lemma 1 hold and when $\|\mathbf{x}\|_\infty = \mathcal{O}(1)$, for $B$-bounded loss $l$, w.h.p. $1 - \delta$, the excess risk for the solution $\hat{\mathbf{w}}_d$ of $\mathrm{PRO}^2$ that uses $d$ features is*

$$\mathcal{L}_T(\hat{\mathbf{w}}_d) - \min_{\mathbf{w} \in \mathcal{W}} \mathcal{L}_T(\mathbf{w}) \| (\boldsymbol{I}_D - \mathsf{M}_d)\mathbf{w}_T^* \|_2 + \frac{\sqrt{d} + B\sqrt{\log(1/\delta)}}{\sqrt{M}}, \tag{12}$$

*where the first term of the RHS controls the bias and the second controls the variance.*

*Proof.* The excess risk for $\hat{\mathbf{w}}_d$ is

$$
\begin{aligned}
&\mathcal{L}_T(\hat{\mathbf{w}}_d) - \min_{\mathbf{w} \in \mathcal{W}} \mathcal{L}_T(\mathbf{w}) \\
&= \mathcal{L}_T(\hat{\mathbf{w}}_d) - \min_{\mathbf{w} \in \mathrm{span}\{\Pi_i\}_{i=1}^d} \mathcal{L}_T(\mathbf{w}) + \min_{\mathbf{w} \in \mathrm{span}\{\Pi_i\}_{i=1}^d} \mathcal{L}_T(\mathbf{w}) - \min_{\mathbf{w} \in \mathcal{W}} \mathcal{L}_T(\mathbf{w}) \\
&= \min_{\mathbf{w} \in \mathrm{span}\{\Pi_i\}_{i=1}^d} \mathcal{L}_T(\mathbf{w}) - \min_{\mathbf{w} \in \mathcal{W}} \mathcal{L}_T(\mathbf{w}) + \mathcal{L}_T(\hat{\mathbf{w}}_d) - \min_{\mathbf{w} \in \mathrm{span}\{\Pi_i\}_{i=1}^d} \mathcal{L}_T(\mathbf{w}) \\
&\| (\boldsymbol{I}_D - \mathsf{M}_d)\mathbf{w}_T^* \|_2 + \frac{\sqrt{d} + B\sqrt{\log(1/\delta)}}{\sqrt{M}}
\end{aligned}
\tag{13}
$$

where the first term is the bias (bounded using Lemma 1), and the second term is the generalization error or the variance (bounded using Lemma 7). $\square$

**Corollary 9.** *Under the SHOG model, $\Pi_1$ recovers the linear discriminant analysis (LDA) solution, $\Pi_1 = \Sigma^{-1}(\mu_2 - \mu_1)/(\|\Sigma^{-1}(\mu_2 - \mu_1)\|_2)$.*

*Proof.* Since the LDA solution is Bayes optimal under the HOG model, it is exactly characterized by the top eigen vector of $\Sigma^{-1}(\mu_2 - \mu_1)(\mu_2 - \mu_1)^\top$. Thus, the Bayes optimal solution on target $\mathbf{w}_T^* \propto \Sigma^{-1}(\mu_2 - \mu_1)$, and since $\Pi_1$ returns the Bayes optimal linear predictor, following Theorem 6, the above corollary is proven. $\square$

**Lemma 10** (bias under SHOG). *Let $\mathsf{M}_d$ be the projection returned by $\mathrm{PRO}^2$. The bias $b_d$ term under our SHOG is $b_d \| (\boldsymbol{I}_D - \boldsymbol{v}_S \boldsymbol{v}_S^\top) \boldsymbol{v}_T \|$. Here, $\boldsymbol{v}_S = \frac{\Sigma_S^{-1} \boldsymbol{\mu}}{\|\Sigma_S^{-1} \boldsymbol{\mu}\|_2}$ and $\boldsymbol{v}_T = \frac{\Sigma_T^{-1} \boldsymbol{\mu}}{\|\Sigma_T^{-1} \boldsymbol{\mu}\|_2}$. Further, when $\|\Sigma_S\|_{\mathrm{op}}$ is bounded, and $\mathsf{M}_d$ is a random rank $d$ projection matrix, $b_d = \mathcal{O}\sqrt{1 - \frac{d}{D}} \cdot \mathrm{KL}(p_S || p_T)$.*

*Proof.* From Corollary 9, we know that $\mathsf{M}_1$ is exactly the rank-1 projection matrix given by the direction $\Sigma_S^{-1}(\mu_2 - \mu_1)/(\|\Sigma_S^{-1}(\mu_2 - \mu_1)\|_2)$. Therefore

$$b_d \leq \|(\boldsymbol{I}_D - \mathsf{M}_d)\mathbf{w}_T^*\|_2 \leq |(\boldsymbol{I}_D - \mathsf{M}_1)\mathbf{w}_T^*\|_2 = \|(\boldsymbol{I}_D - \boldsymbol{v}_S\boldsymbol{v}_S^\top)\boldsymbol{v}_T\|. \tag{14}$$

This gives us the first result for $\boldsymbol{v}_S, \boldsymbol{v}_T$.

For the second result, we note that the KL divergence between multivariate Gaussian distributions is convex.

$$\begin{aligned}
\mathrm{KL}(p_S\|p_T) &= \mathrm{KL}(p(\mathbf{y})p_S(\mathbf{x}\mid\mathbf{y})\|p(\mathbf{y})p_T(\mathbf{x}\mid\mathbf{y})) \\
&\leq \mathrm{KL}(p_S(\mathbf{x}\mid\mathbf{y})\|p_T(\mathbf{x}\mid\mathbf{y})) \\
&= 0.5 \cdot \mathrm{KL}(\mathcal{N}(\mu_1, \Sigma_S)\|\mathcal{N}(\mu_1, \Sigma_T)) + 0.5 \cdot \mathrm{KL}(\mathcal{N}(\mu_2, \Sigma_S)\|\mathcal{N}(\mu_2, \Sigma_T)) \\
&= \frac{1}{2}\mathrm{tr}(\Sigma_T^{-1}\Sigma_S) - \sum_{i=1}^{D}\log\lambda_i^S + \sum_{i=1}^{D}\log\lambda_i^T - D.
\end{aligned} \tag{15}$$

Refer to Wainwright (2019) for the final step, where $\lambda_i^S$ and $\lambda_i^T$ are the eigenvalues of source and target covariance matrices, respectively. The final term in the above derivation is $\mathcal{O}(\mathrm{tr}(\Sigma_T^{-1}))$ when $\Sigma_S = O(1)$. From Lemma 1 we know that under random projections onto $d$ dimensions,

$$b_d \leq L \cdot \sqrt{1 - (d/D)}\|\mathbf{w}_T^*\|\sqrt{1 - (d/D)}\|\Sigma_T^{-1}(\mu_2 - \mu_1)\|\mathrm{tr}(\Sigma_T^{-1}) \tag{16}$$

where we use Corollary 9. Thus from (16) and (15), we get our desired bound:

$$b_d\sqrt{1 - \frac{d}{D}} \cdot \mathrm{KL}(p_S\|p_T).$$

$\square$

**Corollary 11** (tradeoff under SHOG, Corollary 3). *Under our SHOG model of shift, and conditions for a random projection $\mathsf{M}_d$ in Lemma 10, the target error $\mathcal{L}_T(\hat{\mathbf{w}}_d)\mathcal{O}\sqrt{1 - \frac{d}{D}} \cdot \mathrm{KL}(p_S\|p_T) + \sqrt{\frac{d}{M}}$, when $\Sigma_T = O(1)$.*

*Proof.* Direct application of the variance result in Lemma 7 and bias result in Lemma 10, using the same technique used to prove Theorem 2. $\square$

# B  EXPERIMENTAL DETAILS

## B.1  PYTORCH PSEUDOCODE FOR THE PROJECTION STEP

Below, we provide PyTorch pseudocode for the projection step of $\mathrm{PRO}^2$ for binary classification datasets.

```python
def learn_feature_space_basis(x, y, num_features):
    projection = torch.nn.Linear(x.shape[1], num_features)
    opt = torch.optim.AdamW(projection.parameters(), lr=0.01,
                                        weight_decay=0.01)
    max_steps = 100
    for i in range(max_steps):
        logits = projection(x)
        loss = F.binary_cross_entropy_with_logits(logits, y, reduction="
                                        none").mean()
        opt.zero_grad()
        loss.backward()
        opt.step()
        # Enforce orthogonality; we're performing projected gradient
                                        descent
        Q, R = torch.linalg.qr(projection.weight.detach().T)
        projection.weight.data = (Q * torch.diag(R)).T
    feature_space = projection.weight.detach().T
    return feature_space
```

### B.2 Additional dataset details

- **4-Way Collages** (Teney et al., 2021). This binary classification dataset consists of 4-way collages of four images per datapoint, one from each of (1) CIFAR, (2) MNIST, (3) Fashion-MNIST, and (4) SVHN. All four image features are completely correlated in the source data, and we consider four target distributions, where only one of the image features are predictive of the label in each target distribution.

- **Waterbirds** (Sagawa et al., 2020). This dataset tasks the model with classifying images of birds as either a waterbird or landbird. The label is spurious correlated with the background of the image, which is either water or land. There are 4,795 training samples, of which 95% of the data follows the spurious correlation. We use the original training set as the source data and evaluate on 3 different target distributions constructed from the original test dataset: (1) Minority, which contains the test data points that do not follow the spurious correlation, (2) Spurious, containing the points that do, and (3) Balanced, which contains an equal number of points from each of the 4 (bird, background) groups.

- **CelebA** (Liu et al., 2015). Similar to Waterbirds, we use the original training set as source data and evaluate on (1) Minority, (2) Spurious, and (3) Balanced target distributions. In our main experiments in Sec. 6, we use target distributions corresponding to the spurious correlation typically used for evaluation (spurious attribute–gender with label–hair color). Below, in Appendix C include additional results on 4 other variants following the settings used in Lee et al. (2022b): (1) CelebA-1 uses slightly open mouth as the label and wearing lipstick as the spurious attribute, (2) CelebA-2 uses attractive as the label and smiling as the spurious attribute, (3) CelebA-3 uses wavy hair as the label and high cheekbones as the spurious attribute, and finally (4) CelebA-4 uses heavy makeup as the label and big lips as the spurious attribute.

- **Camelyon17** (Bandi et al., 2018). This dataset is part of the WILDS benchmark Koh et al. (2021) and contains medical images where variations in data collection from different hospitals induce naturally occurring distribution shifts. We evaluate on 2 target distributions: (1) ID-Test: a held out test set of images from the source distribution, and (2) OOD-Test: the actual test distribution with a distribution shift due to evaluating data from a different hospital.

- **Living17** (Santurkar et al., 2020). The task is to classify images into one of 17 animal categorie. This dataset presents a subpopulation shift, in that while the ID and OOD distributions have the same overall classes, they contain different subpopulations. We test on the given test set.

- **FMoW** (Koh et al., 2021). This dataset contains satellite images from 5 geographic regions, and the task is the classify the image as one of 62 building or land use types. For the target distribution, we use the subset of the OOD test data belonging to the Africa region.

**Pre-trained models and additional training details.** We extract penultimate embeddings of all source and target datapoints from a pre-trained backbone. We preprocess all datapoints according to the augmentation used during pre-training, and obtain feature embeddings with eval-mode batch normalization. We cache all embeddings for a (backbone, dataset) pair to a single file and train our linear models from the cached file. We use CLIP-ViT-L/16 Dosovitskiy et al. (2020) in our main experiments, and additionally experiment with ResNet18 He et al. (2016), ResNet50, ResNet50-SWaV Caron et al. (2020), CLIP-ViT-B/16 models in Appendix C.3. All pretrained models are publicly available online. We train all models using the AdamW optimizer Loshchilov and Hutter (2017) with weight decay 0.01. For all experiments, we perform early stopping with accuracy on held-out target data and report mean and standard deviation across 10 runs.

## C Additional Experimental Results

### C.1 Additional visualizations for synthetic Gaussian experiment

In Fig. 6, we approximate the bias and variance in the synthetic HOG experiment studied in Fig. 3. On the left, for each test distribution (ID, Near OOD, and Far OOD), we plot the relationship between approximate bias (using error at the largest target dataset size) and nullspace norm and find that they have a roughly linear relationship. Thus, this plot empirically supports the connection supported in the theory between bias and the number of features used, as the nullspace norm decreases as the dimension of the feature-space basis increases.

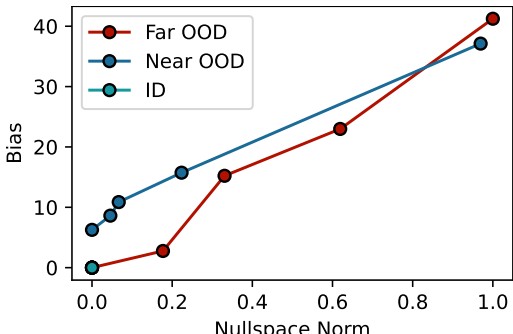 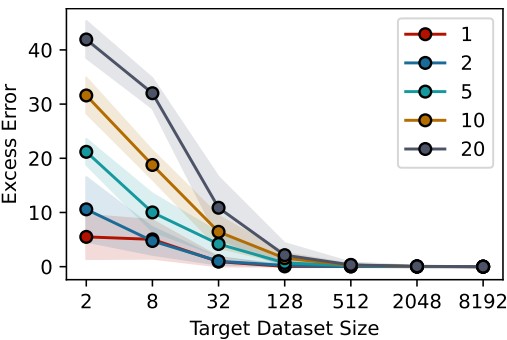

Figure 6: Visualization of bias and variance in the synthetic homoscedastic Gaussian experiment Fig. 3. (Left) We approximate bias by the error at the largest target dataset size, and compare to the nullspace norm. The two quantities have a roughly linear relationship. (Right) We approximate variance by the difference between the error at each dataset size and the error at the largest. We report the average across the three test distributions. Note on the left plot, ID is easily learned and so the corresponding line is therefore clustered near (0, 0), as the nullspace norm and bias are both near 0.

## C.2 EMPIRICAL ANALYSIS OF PROJECTED FEATURE SPACE

We begin by observing the empirical properties of the projected feature space learned during the first projection phase of $\text{PRO}^2$. The Waterbirds dataset consists of "spurious" groups where the background type (land or water) correlates with the bird type (land or water), on which using a shortcut feature that relies on background type will perform optimally, as well as "minority" groups in which the correlation does not hold and requires a robust feature that focuses on the bird itself. On this dataset, we first extract oracle shortcut and robust features by minimizing loss on spurious and minority groups on target data, respectively. These two directions serve as proxies for the optimal classifier on two different target distributions. In addition to $\text{PRO}^2$, we also evaluate a random feature extraction method, which simply samples a random orthonormal basis for the original $D$ embedding space. We plot the nullspace norm of these two features in the subspace spanned by the first $k$ directions, for $1 \le k \le D = 1024$ in **??**. As expected, we see that the earlier features learned by $\text{PRO}^2$ are more similar to the shortcut feature than the robust feature. Because the orthogonality constraint forces the features to be different from each other, the nullspace norm reduces to zero at the highest value $k = 1024$. This experiment shows that the basis learned by $\text{PRO}^2$ contains both the robust and shortcut features for this dataset, and that the robust and shortcut features emerge even for very low-rank bases (i.e., for small values of $d$). In contrast, a random orthogonal basis only captures these two predictive features when the rank is larger. This indicates that our orthogonal projection approach quickly picks up on the most important directions in feature space, which in this case correspond to the shortcut feature representing the background and the robust feature representing the type of bird, as discussed in prior work (Sagawa et al., 2020).

## C.3 USING VARIOUS PRETRAINED BACKBONES

Finally, as $\text{PRO}^2$ relies on using a pre-trained backbone model that is not fine-tuned to initially extract features, we study how different backbones affect performance. In Fig. 7, we plot the accuracy of $\text{PRO}^2$ using 5 pre-trained backbone models that achieve a range of Image-Net accuracies. We find that $\text{PRO}^2$ improves significantly with better pre-trained backbones. These experiments demonstrate the promise of the $\text{PRO}^2$ framework. The quality of pre-trained feature extractors will continue to improve with future datasets and architectures, and $\text{PRO}^2$ leverages such pre-trained backbone models for distribution-shift adaptation in a computationally efficient manner.

## C.4 ABLATION ON THE IMPORTANCE OF ENFORCING ORTHOGONALITY

For the purposes of our empirical analysis, we additionally consider a simpler variant that optimizes the projection matrix $\Pi$ with **N**o **C**onstraint on orthogonality:

$$\Pi_i = \arg\min \mathbb{E}_{(x,y)\sim\mathcal{D}_S}\mathcal{L}(\Pi_i(f(x)), y). \tag{PRO$^2$-NC}$$

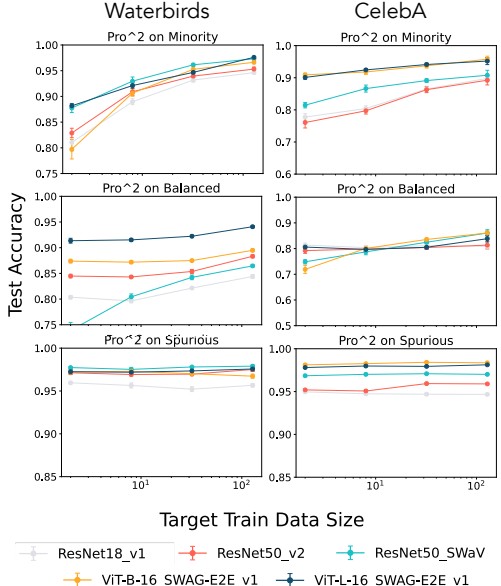

Figure 7: **Different backbones.** We show the accuracy of PRO², where we use various pretrained backbones, which are not fine-tuned. PRO² is able to leverage improvements in the backbone with minimal computational overhead.

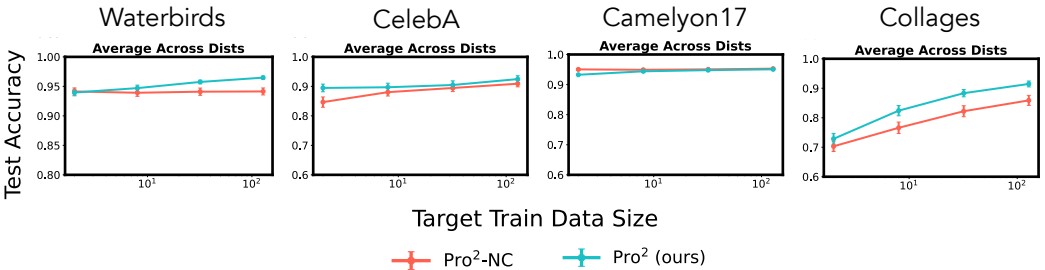

Figure 8: **Importance of orthogonality.** We show the adaptation accuracy of PRO² compared to PRO²-NC, a variant without orthogonality enforced, averaged across the varying target distributions for each dataset.

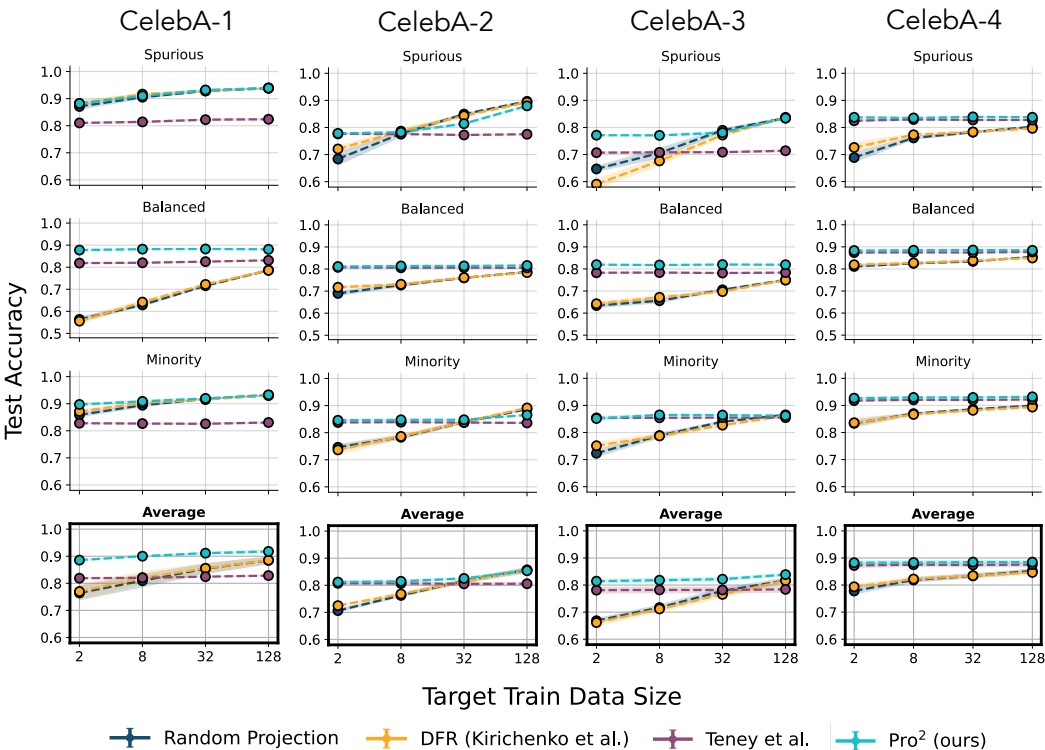

Figure 9: **Main results on additional CelebA variants.** We compare 4 different methods for learning features to adapt to a target distribution: (1) Random Projection, (2) DFR Kirichenko et al. (2022), i.e. standard linear probing, (3) Teney et al. (2021), and (4) PRO². We report target accuracies after probing with different target dataset sizes; we report mean and standard deviation across 10 runs. Similar to the trends seen in Fig. 4, PRO² achieves high accuracy in the low-data regime, substantially outperforming both random orthogonal projection and no projection in most target distributions on all four datasets.

We compare PRO² to PRO²-NC in Fig. 8. While PRO²-NC is is sufficient in some scenarios with milder distribution shift, where the shortcut feature continues to be informative, it fails to learn a diverse set of predictive features and often only learns shortcut features, often failing on more severe distribution shifts.

### C.5    EVALUATION ON ADDITIONAL CELEBA VARIANTS

Finally, in Fig. 9 we supplement our main results in Fig. 4 with additional results from 4 additional variants of CelebA. The takeaways from these results line up with those from Fig. 4. In the few-shot adaptation problem setting, PRO² is consistently the most effective, compared to Random Projection, DFR Kirichenko et al. (2022), which uses standard linear probing, and Teney et al. (2021).

### C.6    ADDITIONAL COMPARISONS

We provide an additional comparison replacing the projection step with PCA (see results below), which performs significantly worse than Pro² on both Waterbirds and CelebA using a range of target data sizes. This is expected since PCA is an unsupervised approach that does not consider labels, whereas Pro² leverages the given labels from the source data in its projection step.

Additionally, we evaluate against two additional points of comparison that use both a small amount of labeled target domain data like Pro², and additionally use unlabeled data: D-BAT (Pagliardini et al., 2022) and DivDis (Lee et al., 2022b), shown in Table 3 While both methods perform well, Pro² at least matches the performance with any target data size and sometimes even outperforms the other two methods, especially with few datapoints (2) on Waterbirds and more datapoints (32) on CelebA, *despite not using any additional unlabeled data*, which may be difficult to obtain.

| | Waterbirds (Balanced) | | | | CelebA (Balanced) | | |
|---|---|---|---|---|---|---|---|
| Method | Target Train Data Size (per label) | | | Method | Target Train Data Size (per label) | | |
| | 2 | 8 | 32 | | 2 | 8 | 32 |
| PCA | 81.5 (1.0) | 85.4 (0.5) | 90.7 (0.3) | PCA | 61.4 (1.8) | 65.6 (0.8) | 75.2 (0.6) |
| Pro$^2$ | 91.8 (0.2) | 91.7 (0.4) | 92.7 (0.3) | Pro$^2$ | 76.5 (1.4) | 78.3 (0.9) | 80.8 (0.9) |

Table 2: **PCA comparison.** PRO$^2$ significantly outperforms PCA by leveraging the source data in its projection step.

| | Waterbirds (Balanced) | | | | CelebA (Balanced) | | |
|---|---|---|---|---|---|---|---|
| Method | Target Train Data Size (per label) | | | Method | Target Train Data Size (per label) | | |
| | 2 | 8 | 32 | | 2 | 8 | 32 |
| D-BAT | 90.1 (0.5) | 91.5 (0.4) | 92.3 (0.2) | D-BAT | 76.7 (0.5) | 78.4 (0.7) | 78.5 (0.7) |
| DivDis | 89.5 (0.8) | 91.1 (0.5) | 93.0 (0.2) | DivDis | 76.2 (0.7) | 77.8 (0.5) | 77.7 (1.0) |
| Pro$^2$ | 91.8 (0.2) | 91.7 (0.4) | 92.7 (0.3) | Pro$^2$ | 76.5 (1.4) | 78.3 (0.9) | 80.8 (0.9) |

Table 3: **Comparison against methods that additionally require unlabeled data.** PRO$^2$ matches and outperforms two state-of-the-art methods that additionally require unlabeled data.

Finally, in Table 4, we provide the zero-shot transfer performance using the same pre-trained backbone (ViT-L-16_SWAG_E2E_v1) trained on the source distribution. We see that using a small amount of target data, PRO$^2$ results in significant gains, particularly on OOD distributions, and thus produces consistently strong performance across a range of test distributions.

| | Waterbirds | | | | CelebA | | |
|---|---|---|---|---|---|---|---|
| | Spurious | Minority | Balanced | | Spurious | Balanced | Minority |
| No Target Data | 97.6 (0.1) | 75.1 (0.4) | 86.1 (0.1) | No Target Data | 97.8 (0.1) | 59.9 (0.4) | 22.6 (0.5) |
| Pro$^2$ | 97.9 (0.2) | 97.0 (0.3) | 94.3 (0.2) | Pro$^2$ | 98.0 (0.1) | 83.7 (0.3) | 96.0 (0.2) |
| | Camelyon17 | | | | Collages | | |
| | ID Test | OOD Test | | | MNIST | CIFAR | Fashion MNIST |
| No Target Data | 93.7 (0.1) | 93.1 (0.0) | | No Target Data | 75.9 (0.1) | 71.8 (0.2) | 61.3 (0.1) |
| Pro$^2$ | 94.5 (0.1) | 93.5 (0.2) | | Pro$^2$ | 98.0 (0.1) | 90.9 (0.3) | 85.0 (0.3) |

Table 4: **Comparison against zero-shot transfer.** PRO$^2$ significantly outperforms the zero-shot transfer performance after training on the source distribution.

## C.7    NUMERICAL VALUES OF EXPERIMENT RESULTS

We include numerical values of experimental results in Figure 4.

| Train Data Size | Method | Balanced | Spurious | Minority |
|---|---|---|---|---|
| 2 | DFR | 82.4 (0.61) | 88.3 (0.88) | 83.2 (0.34) |
| 2 | Pro^2 | 94.1 (0.16) | 97.9 (0.12) | 91.0 (0.30) |
| 2 | Random Projection | 82.9 (0.92) | 88.4 (0.70) | 88.1 (0.61) |
| 8 | DFR | 86.8 (0.53) | 92.3 (0.15) | 91.5 (0.19) |
| 8 | Pro^2 | 94.2 (0.11) | 98.0 (0.19) | 91.2 (0.37) |
| 8 | Random Projection | 87.8 (0.54) | 90.0 (0.75) | 90.7 (0.54) |
| 32 | DFR | 90.4 (0.57) | 94.6 (0.13) | 94.9 (0.29) |
| 32 | Pro^2 | 94.1 (0.11) | 98.2 (0.11) | 94.5 (0.24) |
| 32 | Random Projection | 92.9 (0.33) | 94.0 (0.50) | 94.4 (0.44) |
| 128 | DFR | 94.0 (0.15) | 97.3 (0.11) | 96.8 (0.32) |
| 128 | Pro^2 | 94.4 (0.17) | 98.2 (0.12) | 96.9 (0.24) |
| 128 | Random Projection | 93.8 (0.15) | 96.9 (0.23) | 96.2 (0.43) |

Table 5: **Numerical values of Waterbirds results.**

| Train Data Size | Method | Balanced | Spurious | Minority |
|---|---|---|---|---|
| 2 | DFR | 60.1 (0.80) | 94.5 (0.36) | 84.8 (0.46) |
| 2 | Pro^2 | 77.4 (0.65) | 97.8 (0.23) | 89.1 (0.51) |
| 2 | Random Projection | 59.2 (0.22) | 92.8 (0.43) | 84.4 (0.49) |
| 8 | DFR | 66.3 (0.38) | 96.8 (0.51) | 89.0 (0.15) |
| 8 | Pro^2 | 78.2 (0.17) | 98.1 (0.71) | 91.3 (0.52) |
| 8 | Random Projection | 65.6 (0.45) | 95.7 (0.58) | 89.3 (0.46) |
| 32 | DFR | 72.3 (0.32) | 97.1 (0.33) | 94.5 (0.21) |
| 32 | Pro^2 | 79.4 (0.18) | 97.9 (0.49) | 93.4 (0.34) |
| 32 | Random Projection | 72.9 (0.31) | 97.3 (0.54) | 94.3 (0.42) |
| 128 | DFR | 78.4 (0.35) | 97.9 (0.13) | 95.9 (0.68) |
| 128 | Pro^2 | 81.8 (0.73) | 98.8 (0.24) | 96.4 (0.16) |
| 128 | Random Projection | 79.2 (0.25) | 98.2 (0.17) | 97.0 (0.33) |

Table 6: **Numerical values of CelebA results.**

