# OpenReview forum: "Project and Probe: Sample-Efficient Adaptation by Interpolating Orthogonal Features"
_ICLR.cc/2024/Conference — ICLR 2024 spotlight_

### Official Review · Reviewer_fuyr · 2023-10-31

**Soundness:** 3 good
**Presentation:** 3 good
**Contribution:** 3 good
**Rating:** 8
**Confidence:** 4

**Summary:**

In this paper, the authors propose PROJECT AND PROBE, a lightweight framework consisting of 2 steps: (1) a projection step that extracts a diverse and predictive feature-space basis and (2) a probing step that interpolates between the projected features to efficiently adapt varying target distributions. The core idea is to ensure orthogonality among each component of the feature vector. Each component is utilized by an identical predictor but contains distinct information. Subsequently, these components are employed for predicting the target data. The proposed approach is supported by a theoretical analysis showing that the proposed approach improves sample efficiency.

**Strengths:**

- The paper is well-written and well-organized.
- Enforcing feature orthogonality is intuitive and seems suitable for learning features that remain invariant to distribution shifts.
- The proposed algorithm outperforms the baselines, especially when the sample size of the target data is relatively small.

**Weaknesses:**

- One major limitation of this work is the fact that the project and probe processes are considered solely in the linear model regime due to the pre-trained feature extraction. Also, the theoretical analysis was conducted with a linear model. how would it  extended to large-scale problems?

- The authors only compare with projection-based baselines. I think comparisons with recent general unsupervised domain adaptation methods are needed.

- While learning diverse/orthogonal features is novel in the context of domain adaptation. There is an active line of research that explores this idea in the standard supervised learning setting, such as [1-7]. I think these methods should be discussed in the related work.


[1] Bansal, N., Chen, X., & Wang, Z. Can we gain more from orthogonality regularizations in training deep networks?. Neurips (2018)

[2] Xie, P.; Singh, A.; and Xing, E. P. . Uncorrelation and evenness: a new diversity-promoting regularizer. ICML (2017)

[3] Xie, B.; Liang, Y.; and Song, L.. Diverse neural network learns true target functions. In Artificial Intelligence and Statistics (2017)

[4] Laakom, F., Raitoharju, J., Iosifidis, A., & Gabbouj, M. WLD-Reg: A Data-dependent Within-layer Diversity Regularizer. AAAI (2023)

[5] Cogswell, M.; Ahmed, F.; Girshick, R. B.; Zitnick, L.; and Batra, D. Reducing Overfitting in Deep Networks by Decorrelating Representations. ICLR (2016)

[6] LLaakom, F., Raitoharju, J., Iosifidis, A., & Gabbouj, . Learning distinct features helps, provably. ECML (2023).

[7] Zbontar, J., Jing, L., Misra, I., LeCun, Y., & Deny, S. (2021, July). Barlow twins: Self-supervised learning via redundancy reduction. ICML (2021)

**Questions:**

See Section above.

---

> ### Author Response · Authors · 2023-11-19
>
> We thank you for your review. We provide answers to individual points below.
>
> > One major limitation of this work is the fact that the project and probe processes are considered solely in the linear model regime due to the pre-trained feature extraction. Also, the theoretical analysis was conducted with a linear model. how would it extended to large-scale problems?
>
> There are two key reasons for focusing on the linear setting. One is that the pre-trained feature extractors are quite good, and training linear heads on top of pre-trained features has been shown to be a powerful approach for a wide range of problems. Furthermore, such a pipeline is computationally cheap to run, as the optimization is linear and does not require fine-tuning the whole backbone.The second is that we are considering the low data regime. With only a small number of datapoints from the target domain, a full feature extractor will overfit and not be able to generalize to new test points. As such, the pre-trained feature extraction and linear project and probe head are key to the method, which is why all the analysis is conducted with both of these. We find that our method does work on challenging real-world tasks, such as FMoW satellite images.
>
> > The authors only compare with projection-based baselines. I think comparisons with recent general unsupervised domain adaptation methods are needed.
>
> The problem setting we are studying is different from unsupervised domain adaptation. Whereas unlabeled DA uses unlabeled target data, our setting uses a small labeled dataset from the target distribution; we give a full description in Section 3. Because such methods do not effectively use labeled target domain data, we do not directly compare to them for fairness.
>
> Instead, we have added two additional points of comparison that use both a small amount of labeled target domain data like Pro^2 and additionally use unlabeled data: Pagliardini et al. [47] and Lee et al [27], shown below. While both methods perform well, Pro^2 at least matches the performance with any target data size and sometimes even outperforms the other two methods, especially with few datapoints (2) on Waterbirds and more datapoints (32) on CelebA, despite not using any additional unlabeled data, which may be difficult to obtain.
>
> | Waterbirds (Balanced) |           | Target Train Data Size (per label) |           |
> |:---------------------:|:---------:|:----------------------------------:|:---------:|
> |        Method         |     2     |                 8                 |    32     |
> |         D-BAT         | 90.1 (0.5)|             91.5 (0.4)             | 92.3 (0.2)|
> |         DivDis        | 89.5 (0.8)|             91.1 (0.5)             | 93.0 (0.2)|
> |         Pro^2         | 91.8 (0.2)|             91.7 (0.4)             | 92.7 (0.3)|
>
> | CelebA (Balanced)    |           | Target Train Data Size (per label) |           |
> |:--------------------:|:---------:|:----------------------------------:|:---------:|
> |       Method         |     2     |                 8                 |    32     |
> |       D-BAT          | 76.7 (0.5)|             78.4 (0.7)             | 78.5 (0.7)|
> |       DivDis         | 76.2 (0.7)|             77.8 (0.5)             | 77.7 (1.0)|
> |       Pro^2          | 76.5 (1.4)|             78.3 (0.9)             | 80.8 (0.9)|
>
>
> > While learning diverse/orthogonal features is novel in the context of domain adaptation. There is an active line of research that explores this idea in the standard supervised learning setting, such as [1-7]. I think these methods should be discussed in the related work.
>
> Thank you for the comment and references. We have added these references and some discussion in the related work section. Prior works have explored learning diverse or orthogonal features in standard supervised learning settings and we show how diversification can lead to more sample-efficient adaptation to distribution shifts.
>
> If you have any remaining concerns, please let us know. If we have addressed your concerns, we kindly ask that you may consider raising your score.

---

> > ### Author Response · Authors · 2023-11-21
> > **Checking in**
> >
> > We wanted to follow up on your review and our response. We are open to discussion if you have any additional questions or concerns, and if not, we kindly ask you to consider raising your score.

---

> > > ### Comment · Reviewer_fuyr · 2023-11-21
> > > **Follow up**
> > >
> > > I thank the authors for the response. While most of my concerns are addressed, I am still not convinced by the comparison only with 2-3 projection-based baselines.
> > >
> > > I have an additional question. I understand that Pro^2 is designed to be suitable for learning with a low amount of data. Is it always the case that it is more suitable than other adaptation methods if the dataset is small regardless of the target data complexity?
> > > I don't see any discussion regarding the potential drawbacks or limitations of the approach.

---

> > > > ### Author Response · Authors · 2023-11-21
> > > >
> > > > Thanks for your engagement with our rebuttal. Your original review called for comparisons to "recent general unsupervised domain adaptation methods" so we provided some additional comparisons in the table above with prior works that additionally use unlabeled data. We are unsure what you mean by "comparison only with 2-3 projection-based baselines" in your reply. Can you give a concrete comparison that you would want to see that would help convince you on top of these and the other baselines that we provided in the original paper? We focus on projection-based approaches since they work well with pre-trained networks and are an easier way to learn diverse subset of features from the pretrained set.
> > > >
> > > > For your additional question, for each dataset, we evaluate Pro^2 on several distribution shifts with varying degrees of distribution shift (see e.g. the rows “Spurious”, “Balanced”, “Minority” in Figures 4 and 5), and it is always the best-performing method in the low-data regime, including when there's a larger shift between the target data and the source data. More complex target distributions generally require more directions and therefore more data to fit. Our analysis on the bias-variance tradeoff expressed in Theorem 2 depends on the dataset size M, so Pro^2 will have smaller benefits in settings with a large amount of target data.
> > > >
> > > > Thanks again for your time and effort on reviewing our work. We are happy to answer any additional questions you have.

---

> > > > > ### Comment · Reviewer_fuyr · 2023-11-22
> > > > >
> > > > > > Can you give a concrete comparison that you would want to see that would help convince you on top of these and the other baselines that we provided in the original paper
> > > > >
> > > > > I don't have something exact in mind. But I understand there is limited prior work on this particular problem setting. Even though unfair, I think it is good to include comparisons with different unsupervised DA methods for a complete Comprehensive analysis.  The authors already included two recent unsupervised approaches, namely D-BAT (Pagliardini et al., 2022) and DivDis (Lee et al., 2022b). If the paper gets accepted, i would recommend adding even more unsupervised approaches only for comparative purposes, as this allows us to assess the true value of using extra labeled data and further motivates the setup the authors are proposing.
> > > > >
> > > > > Overall, the authors addressed most of my concerns. I will increase my score.

---

### Official Review · Reviewer_Ft2e · 2023-10-31

**Soundness:** 3 good
**Presentation:** 3 good
**Contribution:** 2 fair
**Rating:** 8
**Confidence:** 3

**Summary:**

The paper introduces Project and Probe (PRO^2), a transfer learning method designed for scenarios with limited target data due to distribution shifts. PRO^2 is based on a two-step approach: first, it projects pre-trained embeddings from the source dataset onto orthogonal directions to derive a diverse, non-redundant set of predictive features; next, it trains a linear classifier on these projected features using the target data. Theoretical analyses emphasize the method's favorable bias-variance tradeoff, and experimental results across four datasets demonstrate an improved performance by 5-15% compared to traditional linear probing methods.

**Strengths:**

- The paper stands out in terms of clarity, organization, and overall presentation. It also offers an extensive appendix that provides in-depth coverage of related topics, adding value for the reader.

- The authors have presented a robust theoretical framework that substantiates their approach. They effectively highlight the method's capability to achieve a desirable balance between bias and variance.

- The empirical experiments are detailed and present a wide range of scenarios. While there are certain reservations (addressed below), the breadth and depth of this section are commendable.

**Weaknesses:**

- The study by Morwani et al. (2023) has previously explored orthogonal projections as a remedy for feature collapse and simplicity bias. This prior exploration somewhat diminishes the uniqueness of the approach presented in this paper.

- Some aspects of the empirical evaluation are unclear and require further details from the authors. See the "Questions" section for more details.

- The paper presents a rather limited set of baselines. Given that the experimental setup seems relatively straightforward, it would be advantageous to have a more comprehensive range of baselines. Specifically, a comparative analysis involving methods anchored in LDA and QDA (as pointed out in Shysheya et al., 2022) could enrich the paper.

**Questions:**

1) Regarding the empirical assessment, was a consistent hyper-parameter search strategy employed across all the evaluated baselines, or was it exclusively used for the proposed model?

2) There are approaches that exploits Linear Discriminant Analysis (LDA) and Quadratic Discriminant Analysis (QDA) for the adaptation of the head of a pretrained model with success (Shysheya et al. 2022). Can the authors comment on the differences between their method and these methods? While the paper covers the relation with respect to LDA, it does not seem to mention the relation with QDA. Adding those baselines to the empirical evaluation may be beneficial.

3) The authors briefly mention the relation with the previous work of Morwani et al. (2023) in the related work section. This section appears somewhat limited and would benefit from a more in-depth exploration. Could the authors elaborate on the parallels and distinctions between their work and Morwani et al. (2023)?

4) Would the authors be able to provide a detailed analysis of the complexity for the proposed method (e.g. FLOPs and/or MACs)? Understanding complexity is crucial as it essentially represents the computational budget. How does the method's time complexity stand in comparison to leading fine-tuning approaches, such as BiT by Kolesnikov et al. (2020), which adapt the entire model body or FiT (Shysheya et al. 2022), which adapt a subset of the body parameters?


References
-----------

Morwani, D., Batra, J., Jain, P., & Netrapalli, P. (2023). Simplicity bias in 1-hidden layer neural networks. arXiv preprint arXiv:2302.00457.

Kolesnikov, A., Beyer, L., Zhai, X., Puigcerver, J., Yung, J., Gelly, S., & Houlsby, N. (2020). Big transfer (bit): General visual representation learning. In Computer Vision–ECCV 2020: 16th European Conference, Glasgow, UK, August 23–28, 2020, Proceedings, Part V 16 (pp. 491-507). Springer International Publishing.

Shysheya, A., Bronskill, J., Patacchiola, M., Nowozin, S., & Turner, R. E. (2022). Fit: Parameter efficient few-shot transfer learning for personalized and federated image classification. arXiv preprint arXiv:2206.08671.

---

> ### Author Response · Authors · 2023-11-19
>
> Thank you for your comments. We provide answers to individual points below.
>
> >  There are approaches that exploits Linear Discriminant Analysis (LDA) and Quadratic Discriminant Analysis (QDA) for the adaptation of the head of a pretrained model with success (Shysheya et al. 2022). Can the authors comment on the differences between their method and these methods? While the paper covers the relation with respect to LDA, it does not seem to mention the relation with QDA. Adding those baselines to the empirical evaluation may be beneficial.
>
> Both LDA and QDA operate in the setting without distribution shift, and are typically used to learn Bayes optimal predictors on in-distribution data samples under assumptions on the class conditional distributions being Gaussians. On the contrary, our method extracts linear classifiers for few shot adaptation on out-of-distribution data. This requires extracting more directions than just the optimal classifiers on in-distribution source data, which we accomplish with our “project” step. Nevertheless, in Corollary 9, we show that Pro^2 with dimensionality d = 1 provably recovers the LDA direction in a shifted homoscedastic Gaussian model. Similarly, under heteroscedastic Gaussian model, minimizing cross entropy loss with a quadratic kernel (feature extractor), will recover the QDA classifier for d = 1. We also show that using higher values of d is critical in adapting to higher degrees of distribution shift. In this sense, the projection of Pro^2 can be seen as a generalization of LDA/QDA. We compare to a wide range of state-of-the-art baselines (e.g. DFR, Teney et al., D-BAT, DivDis, etc) and include a comparison between our method and replacing the projection step with PCA in the Appendix.
>
> > Regarding the empirical assessment, was a consistent hyper-parameter search strategy employed across all the evaluated baselines, or was it exclusively used for the proposed model?
>
> As stated in Section 6.1, we employ the same hyperparameter search strategy across all the evaluated baselines. For all comparisons, we tune hyperparameters over 3 different learning rates (0.1, 0.01, and 0.001) as well as 3 different L2 regularization weights (0.1, 0.01, 0.001). We also sweep over 6 different projection dimensions (d = 1, 4, 16, 64, 256, 1024) and report results over 10 runs.
>
> > The authors briefly mention the relation with the previous work of Morwani et al. (2023) in the related work section. This section appears somewhat limited and would benefit from a more in-depth exploration. Could the authors elaborate on the parallels and distinctions between their work and Morwani et al. (2023)?
>
> We apologize for any confusion. Morwani et al (2023) is a related concurrent work that studies the simplicity bias of 1-hidden layer networks. Their work and ours are related in that we are both studying the tendency of neural networks to rely on the simplest low-dimensional projection of inputs that fit the training data. Their theory leverages the literature on the infinite-width limit of 1-layer networks, whereas we directly study the bias-variance tradeoff in the finite-width setting using tools from information theory. Our understanding of the bias-variance tradeoff naturally suggests our method, which interpolates among orthogonal features and shows improved sample efficiency in practice.
>
> > Would the authors be able to provide a detailed analysis of the complexity for the proposed method (e.g. FLOPs and/or MACs)? Understanding complexity is crucial as it essentially represents the computational budget. How does the method's time complexity stand in comparison to leading fine-tuning approaches, such as BiT by Kolesnikov et al. (2020), which adapt the entire model body or FiT (Shysheya et al. 2022), which adapt a subset of the body parameters?
>
> Our training involves at most two linear layers on top of cached feature vectors and is therefore generally much more computationally efficient than methods that adapt the entire model body or a large subset. The optimization problem is linear, so we were able to perform 30k runs within 24 hours using four standard CPUs and no GPUs.
>
> We are happy to answer any further questions you may have. If we have addressed your concerns, we kindly ask that you may consider raising your score.

---

> > ### Author Response · Authors · 2023-11-21
> > **Checking in**
> >
> > We wanted to follow up on your review and our response. We are open to discussion if you have any additional questions or concerns, and if not, we kindly ask you to consider raising your score.

---

> > > ### Comment · Reviewer_Ft2e · 2023-11-22
> > > **Answer to rebuttal**
> > >
> > > Thank you for the detailed answer and the clarifications. I do not have any further questions, I will update my score.

---

### Official Review · Reviewer_sNr1 · 2023-11-01

**Soundness:** 3 good
**Presentation:** 3 good
**Contribution:** 2 fair
**Rating:** 5
**Confidence:** 4

**Summary:**

This paper deals with the problem of transfer learning with a small amount of target data. It proposes Project and Probe, which first learns a liner projection that maps the pre-trained embedding onto orthogonal directions and then learns a linear classifier on top of the projected features on the small target dataset. The proposed method outperforms prior methods on transfer learning when given very limited target data.

**Strengths:**

- The paper is clearly written and organized.

- The enhanced sample-efficient generalization of the proposed method is supported by theoretical analysis.

**Weaknesses:**

- A critical comparison is missing from the experiments: How does the proposed method perform compared to zero-shot transfer learning methods (i.e., no target training data), as cited in related work?

- For reproducibility, it is necessary to include the numerical values of the experimental results in addition to the line charts.

**Questions:**

See weaknesses.

---

> ### Author Response · Authors · 2023-11-19
>
> Thank you for your review. In light of our responses to your individual points, please let us know if you have any remaining concerns.
>
> > A critical comparison is missing from the experiments: How does the proposed method perform compared to zero-shot transfer learning methods (i.e., no target training data), as cited in related work?
>
> We provide the zero-shot transfer performance using the same pre-trained backbone (ViT-L-16_SWAG_E2E_v1) of the datasets below.
>
> |                     | Waterbirds            |             |             | CelebA                |             |             |
> |---------------------|-----------------------|-------------|-------------|-----------------------|-------------|-------------|
> |                     | Spurious              | Balanced    | Minority   | Spurious              | Balanced    | Minority    |
> | No Target Data      | 97.6 (0.1)            | 86.1 (0.1)  | 75.1 (0.4)  | No Target Data        | 97.8 (0.1)  | 59.9 (0.4)  | 22.6 (0.5) |
> | Pro^2               | 97.9 (0.2)            | 94.3 (0.2)  | 97.0 (0.3)  | Pro^2                 | 98.0 (0.1)  | 83.7 (0.3)  | 96.0 (0.2) |
>
> |                     | Camelyon17           |             |             | Collages              |             |             |            |
> |---------------------|----------------------|-------------|-------------|-----------------------|-------------|-------------|------------|
> |                     | ID Test              | OOD Test    |             | Collages-MNIST        | Collages-CIFAR | Collages-Fashion MNIST |
> | No Target Data      | 93.7 (0.1)           | 93.1 (0.0)  | No Target Data | 75.9 (0.1)         | 71.8 (0.2)    | 61.3 (0.1)            |
> | Pro^2               | 94.5 (0.1)           | 93.5 (0.2)  | Pro^2        | 98.0 (0.1)           | 90.9 (0.3)    | 85.0 (0.3)            |
>
> We see that using a small amount of target data, Pro^2 results in significant gains, particularly on OOD distributions, and thus produces consistently strong performance across a range of test distributions.
>
> > For reproducibility, it is necessary to include the numerical values of the experimental results in addition to the line charts.
>
> We have added numerical values of the experimental results to the end of the Appendix in the revised pdf.
>
> We hope that our response has addressed all your questions and concerns. We kindly ask you to let us know if you have any remaining concerns, and - if we have answered your questions - to reevaluate your score.

---

> > ### Author Response · Authors · 2023-11-21
> > **Checking in**
> >
> > We wanted to follow up on your review and our response. We are open to discussion if you have any additional questions or concerns, and if not, we kindly ask you to reevaluate your score and assessment of our work.

---

> > > ### Author Response · Authors · 2023-11-22
> > > **Following up**
> > >
> > > Thanks again for your review. We wanted to follow up again to make sure that your concerns are being properly addressed. Please let us know if you have additional questions. if all your concerns have been resolved, we would greatly appreciate it if you could reconsider and adjust your rating and evaluation of our work.

---

### Public Comment · ~Grzegorz_Rypeść1 · 2024-05-08
**Reproducibility**

Hello, could you please share the code to reproduce the results? I cannot find the code in supplementary materials nor on public repositories. Thanks!

---

### Meta-Review · Area_Chair_APh5 · 2023-12-18

**Metareview:**

The paper proposed project and probe method for dealing with ood transfer learning. The code idea is clean and effective, and the reviewers have largely held postive views of the response. The author response addressed most concerns that arose in the initial reviews leading 2/3 engaged authors to increase the scores. Overall the meta reviewer believes the paper introduces a simple idea and demonstrates its usefulness effectively which could be impactful in the community.

**Justification For Why Not Higher Score:**

I believe the paper is valuable contribution in understanding OOD. However, the meta reviewer feels the restriction to limited types of OOD datasets makes the idea not as impactful as an oral paper

**Justification For Why Not Lower Score:**

The reviewers largely held positive reviews. Further, simple ideas that can help with adaptation for transfer learning/continual learning in new distributions are useful for the community to explore. I would recommend accept confidently, but will be ok if the acceptance was rated down to poster.

---

### Decision · Program_Chairs · 2024-01-16

Accept (spotlight)